# The Generalized Lasso with Nonlinear Observations and Generative Priors

**Zhaoqiang Liu**
National University of Singapore
dcslizha@nus.edu.sg

**Jonathan Scarlett**
National University of Singapore
scarlett@comp.nus.edu.sg

## Abstract

In this paper, we study the problem of signal estimation from noisy non-linear measurements when the unknown $n$-dimensional signal is in the range of an $L$-Lipschitz continuous generative model with bounded $k$-dimensional inputs. We make the assumption of sub-Gaussian measurements, which is satisfied by a wide range of measurement models, such as linear, logistic, 1-bit, and other quantized models. In addition, we consider the impact of adversarial corruptions on these measurements. Our analysis is based on a generalized Lasso approach (Plan and Vershynin, 2016). We first provide a non-uniform recovery guarantee, which states that under i.i.d. Gaussian measurements, roughly $O\left(\frac{k}{\epsilon^2}\log L\right)$ samples suffice for recovery with an $\ell_2$-error of $\epsilon$, and that this scheme is robust to adversarial noise. Then, we apply this result to neural network generative models, and discuss various extensions to other models and non-i.i.d. measurements. Moreover, we show that our result can be extended to the uniform recovery guarantee under the assumption of a so-called local embedding property, which is satisfied by the 1-bit and censored Tobit models.

## 1 Introduction

In standard compressive sensing (CS) [9, 37], one considers a linear observation model of the form

$$y_i = \langle \mathbf{a}_i, \mathbf{x}^* \rangle + \epsilon_i, \quad i = 1, 2, \ldots, m, \tag{1}$$

where $\mathbf{x}^* \in \mathbb{R}^n$ is an unknown $k$-sparse signal vector, $\mathbf{a}_i \in \mathbb{R}^n$ is the $i$-th measurement vector, and $\epsilon_i \in \mathbb{R}$ is the noise term. The goal is to accurately recover $\mathbf{x}^*$ given $\mathbf{A}$ and $\mathbf{y}$, where $\mathbf{A} \in \mathbb{R}^{m \times n}$ is the measurement matrix whose $i$-th row is $\mathbf{a}_i^T$, and $\mathbf{y} \in \mathbb{R}^m$ is the vector of observations. To obtain an estimate of $\mathbf{x}^*$, a natural idea is to minimize the $\ell_2$ loss subject to a structural constraint:

$$\text{minimize } \|\mathbf{A}\mathbf{x} - \mathbf{y}\|_2 \quad \text{subject to} \quad \mathbf{x} \in \mathcal{K}, \tag{2}$$

where $\mathcal{K}$ captures the structure of $\mathbf{x}^*$; this may be set to be the set of all $k$-sparse vectors in $\mathbb{R}^n$, or for computational reasons, may instead be the scaled $\ell_1$-ball, giving rise to the constrained Lasso [33]. We refer to (2) as the $\mathcal{K}$-Lasso (with observations $\mathbf{y}$ and measurement matrix $\mathbf{A}$).

Despite the far-reaching utility of standard CS, in many real-world applications the assumption of a linear model is too restrictive. To address this problem, the semi-parametric *single index model (SIM)* is considered in various papers [10, 25, 26]:

$$y_i = f(\langle \mathbf{a}_i, \mathbf{x}^* \rangle), \quad i = 1, 2, \ldots, m, \tag{3}$$

where $f : \mathbb{R} \to \mathbb{R}$ is an unknown (possibly random) function that is independent of $\mathbf{a}_i$. In general, $f$ plays the role of a nonlinearity, and we aim to estimate the signal $\mathbf{x}^*$ despite this unknown nonlinearity. Note that the norm of $\mathbf{x}^*$ is sacrificed in SIM, since it may be absorbed into the unknown function $f$. Hence, for simplicity of presentation, we assume that $\mathbf{x}^*$ is a unit vector in $\mathbb{R}^n$. In this paper, similar

to that in [26], we make the assumption that the random variables $y_i$ are sub-Gaussian; see Section 2.2 for several examples. In addition, to further strengthen the robustness guarantees, we also allow for adversarial noise. That is, we consider the case of corrupted observations $\tilde{\mathbf{y}}$ that can be produced from $\mathbf{y}$ in an arbitrary manner (possibly depending on $\mathbf{A}$) subject to an $\ell_2$-norm constraint.

Motivated by the tremendous success of deep generative models in abundance of real applications [8], a new perspective of CS has recently emerged, in which the assumption that the underlying signal can be well-modeled by a (deep) generative model replaces the common sparsity assumption [2]. In addition to the theoretical developments, existing works have presented impressive numerical results for CS with generative models, with large reductions (e.g., a factor of 5 to 10) in the required number of measurements compared to sparsity-based methods [2].

## 1.1 Related Work

In this subsection, we provide a summary of some relevant existing works. These works can roughly be divided into (i) CS with generative models, and (ii) SIM without generative models.

**CS with generative models:** Bora *et al.* [2] show that for an $L$-Lipschitz continuous generative model with bounded $k$-dimensional inputs, roughly $O(k \log L)$ random Gaussian linear measurements suffice for an accurate recovery. The analysis in [2] is based on the $\mathcal{K}$-Lasso (2), as well as showing that a natural counterpart to the Restricted Eigenvalue Condition (REC), termed the Set-REC (S-REC), is satisfied by Gaussian measurement matrices. Extensive experimental results for the $\mathcal{K}$-Lasso have been presented in [2] in the case of linear measurements. Follow-up works of [2] provide certain additional algorithmic guarantees [5,13,23,28], as well as information-theoretic lower bounds [15,18].

1-bit CS with generative models has been studied in various recent works [17,27]. In [27], the authors study robust 1-bit recovery for $d$-layer, $w$-width ReLU neural network generative models, and the dithering technique [6,14,16,40] is used to enable the estimation of the norm. It is shown that roughly $O\left(\frac{kd}{\epsilon^2} \log w\right)$ sub-exponential measurements guarantee the uniform recovery[1] of any signal in the range of the generative model up to an $\ell_2$-error of $\epsilon$. These results do not apply to general non-linear measurement models, and the authors only consider ReLU neural networks with no offsets, rather than general deep generative models. In addition, the algorithm analyzed is different to the $\mathcal{K}$-Lasso.

The authors of [17] prove that the so-called Binary $\epsilon$-Stable Embedding property (B$\epsilon$SE) holds for 1-bit compressive sensing with $L$-Lipschitz continuous generative models with $O\left(\frac{k}{\epsilon^2} \log \frac{L}{\epsilon^2}\right)$ Gaussian measurements. However, these theoretical results are information-theoretic in nature, and computationally efficient algorithms are not considered. Hence, when specialized to the 1-bit setting, our results complement those of [17] by considering the $\mathcal{K}$-Lasso, which can be approximated efficiently via gradient descent [2].

The work [39] is perhaps closest to our work, and considers the estimation of a signal in the range of an $L$-Lipschitz continuous generative model from non-linear and heavy-tailed measurements. By considering estimators via score functions based on the first and second order Stein's identity, it is shown that roughly $O\left(\frac{k}{\epsilon^2} \log L\right)$ measurements suffice for achieving non-uniform recovery with $\ell_2$-error at most $\epsilon$. The authors make the assumption that the nonlinearity $f$ is *differentiable*, which fails to hold in several cases of interest (e.g., 1-bit and other quantized measurements). In addition, the estimators based on the above-mentioned score functions differ significantly from the $\mathcal{K}$-Lasso.

**SIM without generative models:** The authors of [26] consider SIM and a measurement matrix with i.i.d. standard Gaussian entries. The analysis is based on the estimates of both the global and local Gaussian mean width (GMW) of the set of structured signals $\mathcal{K}$, which can be used to understand its effective dimension. The work [25] generalizes the results in [26] to allow $\mathbf{a}_i \sim \mathcal{N}(\mathbf{0}, \boldsymbol{\Sigma})$ with an unknown covariance matrix $\boldsymbol{\Sigma}$, derives tighter results when specialized to the linear model. In these papers, $\mathcal{K}$ is assumed to be star-shaped[2] or convex, which may not be satisfied for the range of general Lipschitz continuous generative models. In addition, without further assumptions on the signal $\mathbf{x}^*$, the structured set $\mathcal{K}$, or the measurement model, the recovery error bound exhibits $m^{-\frac{1}{4}}$ scaling, which is weaker than the typical $m^{-\frac{1}{2}}$ scaling. In each of these papers, only non-uniform recovery guarantees are provided. See the table in Appendix A for a more detailed overview.

Further follow-up works of [25, 26] include [10] and [20]. In [10], the results for the $\mathcal{K}$-Lasso are extended to a fairly large class of convex loss functions with the assumption that $\mathcal{K}$ is convex. The authors of [20] develop a framework for characterizing time-data tradeoffs for various algorithms used to recover a structured signal from nonlinear observations.

For high-dimensional SIM with heavy-tailed elliptical symmetric measurements, [12, 38] propose thresholded least square estimators to attain similar performance guarantees to those for the Gaussian case. Thresholded score function estimators via Stein's identity are proposed in [41, 42], with the purpose of obtaining a consistent estimator for general non-Gaussian measurements. While treating heavy-tailed measurements, their methods depend heavily on the chosen basis, and appear difficult to generalize beyond sparse and low-rank signals.

Sharp error bounds (including constants) for generalized Lasso problems were provided in [21, 22, 30, 31]. Our focus is on scaling laws, and we leave refined studies of this kind for future work.

A table comparing our results with the most relevant existing works is given in Appendix A.

## 1.2 Contributions

In this paper, we provide recovery guarantees for the $\mathcal{K}$-Lasso with non-linear and corrupted observations under a generative prior. Our main results are outlined as follows:

- In Section 3, we characterize the number of measurements sufficient to attain a non-uniform and accurate recovery of an underlying signal in the range of a Lipschitz continuous generative model. In Section 3.2, we specialize this result to neural network generative models.
- In Section 4, we discuss several variations or extensions of our main result, including an unknown covariance matrix for the random measurement vectors, relaxing the norm restriction for the underlying signal, and considering bounded $k$-sparse vectors.
- In Section 5, we provide uniform recovery guarantees under the assumption of a local embedding property, which holds for various models of interest.

## 1.3 Notation

We use upper and lower case boldface letters to denote matrices and vectors respectively. We write $[N] = \{1, 2, \cdots, N\}$ for a positive integer $N$. A *generative model* is a function $G : \mathcal{D} \to \mathbb{R}^n$, with latent dimension $k$, ambient dimension $n$, and input domain $\mathcal{D} \subseteq \mathbb{R}^k$. For a set $S \subseteq \mathbb{R}^k$ and a generative model $G : \mathbb{R}^k \to \mathbb{R}^n$, we write $G(S) = \{G(\mathbf{z}) : \mathbf{z} \in S\}$. We use $\|\mathbf{X}\|_{2\to2}$ to denote the spectral norm of a matrix $\mathbf{X}$. We define the $\ell_q$-ball $B_q^k(r) := \{\mathbf{z} \in \mathbb{R}^k : \|\mathbf{z}\|_q \leq r\}$ for $q \in [0, +\infty]$, and we use $B_q^k$ to abbreviate $B_q^k(1)$. $\mathcal{S}^{n-1} := \{\mathbf{x} \in \mathbb{R}^n : \|\mathbf{x}\|_2 = 1\}$ represents the unit sphere in $\mathbb{R}^n$. The symbols $C, C', C'', c, c'$ are absolute constants whose values may differ from line to line.

# 2 Problem Setup

In this section, we formally introduce the problem, and overview the main assumptions that we adopt. In addition, we provide examples of measurement models satisfying our assumptions.

Before proceeding, we state the following standard definition.

**Definition 1.** *A random variable $X$ is said to be sub-Gaussian if there exists a positive constant $C$ such that $(\mathbb{E}[|X|^p])^{1/p} \leq C\sqrt{p}$ for all $p \geq 1$. The sub-Gaussian norm of a sub-Gaussian random variable $X$ is defined as $\|X\|_{\psi_2} := \sup_{p\geq 1} p^{-1/2} (\mathbb{E}[|X|^p])^{1/p}$.*

## 2.1 Setup and Main Assumptions

Recall that the (uncorrupted) measurement model is given in (3), where the function $f(\cdot)$ may be random (but independent of $\mathbf{A}$). Except where stated otherwise, we make the following assumptions:

- The measurement matrix $\mathbf{A}$ has i.i.d. standard Gaussian entries, i.e., $\mathbf{a}_i \overset{i.i.d.}{\sim} \mathcal{N}(\mathbf{0}, \mathbf{I}_n)$.
- The scaled vector $\mu\mathbf{x}^*$ lies in a set of structured signals $\mathcal{K}$, where $\mu$ is a fixed parameter depending on $f$ specified below. We focus on the case that $\mathcal{K} = \mathrm{Range}(G)$ for some $L$-Lipschitz continuous generative model $G : B_2^k(r) \to \mathbb{R}^n$ (e.g., see [2]).
- Similarly to [11, 26, 29], we assume that each $y_i$ is (unconditionally) sub-Gaussian.

- In addition to any random noise in $f$, we allow for adversarial noise. In this case, instead of observing $\mathbf{y}$ directly, we only assume access to $\tilde{\mathbf{y}} = [\tilde{y}_1, \ldots, \tilde{y}_m]^T \in \mathbb{R}^m$ satisfying

$$\frac{1}{\sqrt{m}} \|\tilde{\mathbf{y}} - \mathbf{y}\|_2 \leq \tau \tag{4}$$

  for some parameter $\tau \geq 0$. Note that the corruptions of $\mathbf{y}$ yielding $\tilde{\mathbf{y}}$ may depend on $\mathbf{A}$.
- To derive an estimate of $\mathbf{x}^*$ (up to constant scaling), we seek $\hat{\mathbf{x}}$ minimizing $\|\tilde{\mathbf{y}} - \mathbf{A}\mathbf{x}\|_2$ over $\mathcal{K}$:

$$\hat{\mathbf{x}} = \arg\min_{\mathbf{x} \in \mathcal{K}} \|\tilde{\mathbf{y}} - \mathbf{A}\mathbf{x}\|_2. \tag{5}$$

  Recall that we refer to this generalized Lasso as the $\mathcal{K}$-Lasso (with corrupted observations $\tilde{\mathbf{y}}$ and measurement matrix $\mathbf{A}$) [25,26]. It may seem counter-intuitive that the $\mathcal{K}$-Lasso is provably accurate even for non-linear observations; the idea is that the nonlinearity is rather treated as noise and one may transform the non-linear observation model into a scaled linear model with an unconventional noise term [25].
- Let $g \sim \mathcal{N}(0,1)$ be a standard normal random variable. To analyze the recovery performance as a function of the nonlinearity $f$, we use the following parameters, which play a key role:
  - The mean term, denoted $\mu := \mathbb{E}[f(g)g]$;
  - The sub-Gaussian norm of $f(g)$ (i.e., of any given $y_i$), denoted $\psi := \|f(g)\|_{\psi_2}$.

**Remark 1.** *For $\mathcal{K} = \mathrm{Range}(G)$, the estimator (5) was considered in [2], focusing on linear observations. It was noted that although finding an exact solution may be difficult due to the (typical) non-convexity of $G$, gradient methods for finding an approximate solution are effective in practice.*

## 2.2 Examples of Measurement Models

When $f$ does not grow faster than linearly, i.e., $|f(x)| \leq a + b|x|$ for some scalars $a$ and $b$, $y_i$ will be sub-Gaussian. We may also consider various nonlinear models that give sub-Gaussian observations. For example, the censored Tobit model, $f(x) = \max\{x, 0\}$, gives $\mu = \frac{1}{2}$ and $\psi \leq C$.

In addition, by setting $a = 1$ and $b = 0$ in the above-mentioned condition $|f(x)| \leq a + b|x|$, we observe that measurement models with each output selected from $\{-1, 1\}$, i.e., 1-bit measurements, lead to sub-Gaussian $y_i$. For example, for 1-bit observations with random bit flips, we set $f(x) = \xi \cdot \mathrm{sign}(x)$, where $\xi$ is an independent $\pm 1$-valued random variable with $\mathbb{P}(\xi = -1) = p < \frac{1}{2}$. In this case, we have $\mu = (1 - 2p)\sqrt{\frac{2}{\pi}}$ and $\psi = 1$ [10].

We may also consider additive noise before 1-bit quantization, i.e., $f(x) := \mathrm{sign}(x + z)$, where $z$ is an independent noise term. Different forms of noise lead to distinct binary statistical models. For example, if $z$ is Gaussian, this corresponds to the probit model. On the other hand, if $z$ is logit noise, this recovers the logistic regression model. More generally, we may consider non-binary quantization schemes, such as uniform (mid-riser) quantization: For some $\Delta > 0$, $f(x) = \Delta \left(\lfloor \frac{x}{\Delta} \rfloor + \frac{1}{2}\right)$. It is easy to see that $|f(x)| \leq |x| + \frac{\Delta}{2}$ for all $x \in \mathbb{R}$, and thus the corresponding observations are sub-Gaussian. In addition, we have $\mu = 1$ and $\psi \leq C + \frac{\Delta}{2}$ [32].

It is also worth noting that certain popular models, such as phase retrieval [3, 7] with $f(x) = x^2$, do not satisfy the sub-Gaussianity assumption. Such models are beyond the scope of this work, but could potentially be handled via analogous ideas to [25].

## 3 Main Result

In the following, we state our main theorem concerning non-uniform recovery, i.e., the vector $\mathbf{x}^*$ is fixed in advance, before the sample matrix $\mathbf{A}$ is drawn.

**Theorem 1.** *Consider any $\mathbf{x}^* \in \mathcal{S}^{n-1} \cap \frac{1}{\mu}\mathcal{K}$ with $\mathcal{K} = G(B_2^k(r))$ for some $L$-Lipschitz $G$ : $B_2^k(r) \to \mathbb{R}^n$, along with $\mathbf{y}$ from the model (3) with $\mathbf{a}_i \overset{i.i.d.}{\sim} \mathcal{N}(\mathbf{0}, \mathbf{I}_n)$, and an arbitrary corrupted vector $\tilde{\mathbf{y}}$ with $\frac{1}{\sqrt{m}}\|\tilde{\mathbf{y}} - \mathbf{y}\|_2 \leq \tau$. For any $\epsilon > 0$, if $Lr = \Omega(\epsilon\psi n)$ and $m = \Omega\left(\frac{k}{\epsilon^2} \log \frac{Lr}{\epsilon\psi}\right),$[3] then with probability $1 - e^{-\Omega(\epsilon^2 m)}$, any solution $\hat{\mathbf{x}}$ to the $\mathcal{K}$-Lasso (5) satisfies*

$$\|\mu\mathbf{x}^* - \hat{\mathbf{x}}\|_2 \leq \psi\epsilon + \tau. \tag{6}$$

If $\tau = 0$ and $\psi > 0$ is fixed, we get an error bound on the order of $\sqrt{\frac{k \log(Lr)}{m}}$ up to a logarithmic factor in $m$, in particular matching the usual $m^{-\frac{1}{2}}$ scaling. In addition, the $k \log(Lr)$ dependence (as well as the effect of $\tau > 0$) is consistent with prior work on the linear [2] and 1-bit [17] models, while also holding for broader non-linear models. Additional variations are given in Section 4, and a uniform guarantee is established in Section 5 under additional assumptions.

### 3.1  Proof of Theorem 1

Before presenting the proof of Theorem 1, we state some useful auxiliary results. First, we present the definition of the Set-Restricted Eigenvalue Condition (S-REC) [2], which generalizes the REC.

**Definition 2.** *Let $S \subseteq \mathbb{R}^n$. For parameters $\gamma > 0$, $\delta \geq 0$, a matrix $\tilde{\mathbf{A}} \in \mathbb{R}^{m \times n}$ is said to satisfy the S-REC($S, \gamma, \delta$) if, for every $\mathbf{x}_1, \mathbf{x}_2 \in S$, it holds that*

$$\|\tilde{\mathbf{A}}(\mathbf{x}_1 - \mathbf{x}_2)\|_2 \geq \gamma \|\mathbf{x}_1 - \mathbf{x}_2\|_2 - \delta. \tag{7}$$

Recalling that $\mathbf{A} \in \mathbb{R}^{m \times n}$ has i.i.d. $\mathcal{N}(0, 1)$ entries, the following lemma from [2] shows that $\frac{1}{\sqrt{m}}\mathbf{A}$ satisfies the S-REC condition for bounded Lipschitz generative models.

**Lemma 1.** ([2, Lemma 4.1]) *Fix $r > 0$, and let $G : B_2^k(r) \to \mathbb{R}^n$ be $L$-Lipschitz. For $\alpha \in (0, 1)$, if $m = \Omega\left(\frac{k}{\alpha^2} \log \frac{Lr}{\delta}\right)$, then a random matrix $\frac{1}{\sqrt{m}}\mathbf{A} \in \mathbb{R}^{m \times n}$ with $a_{ij} \overset{i.i.d.}{\sim} \mathcal{N}(0, 1)$ satisfies the S-REC($G(B_2^k(r), 1 - \alpha, \delta)$) with probability $1 - e^{-\Omega(\alpha^2 m)}$.*

Using a basic two-sided concentration bound for standard Gaussian matrices (*cf.*, Lemma 4 in Appendix B), and a simple modification of the proof of Lemma 1 given in [2], we obtain the following lemma, which is useful for upper bounding the error corresponding to adversarial noise.

**Lemma 2.** *Fix $r > 0$, and let $G : B_2^k(r) \to \mathbb{R}^n$ be $L$-Lipschitz. For $\alpha < 1$ and $\delta > 0$, if $m = \Omega\left(\frac{k}{\alpha^2} \log \frac{Lr}{\delta}\right)$, then with probability $1 - e^{-\Omega(\alpha^2 m)}$, we have for all $\mathbf{x}_1, \mathbf{x}_2 \in G(B_2^k(r))$ that*

$$\frac{1}{\sqrt{m}}\|\mathbf{A}\mathbf{x}_1 - \mathbf{A}\mathbf{x}_2\|_2 \leq (1 + \alpha)\|\mathbf{x}_1 - \mathbf{x}_2\|_2 + \delta. \tag{8}$$

Using a chaining argument similar to [2, 17], we additionally establish the following technical lemma, whose proof is given in Appendix B. Note that here we only require that $\tilde{\mathbf{x}} \in \mathcal{K} = G(B_2^k(r))$, and we do not require that $\mu\bar{\mathbf{x}} \in \mathcal{K}$.

**Lemma 3.** *Fix any $\bar{\mathbf{x}} \in \mathcal{S}^{n-1}$ and let $\bar{\mathbf{y}} := f(\mathbf{A}\bar{\mathbf{x}})$. Suppose that some $\tilde{\mathbf{x}} \in \mathcal{K}$ is selected depending on $\bar{\mathbf{y}}$ and $\mathbf{A}$.[4] For any $\delta > 0$, if $Lr = \Omega(\delta n)$ and $m = \Omega\left(k \log \frac{Lr}{\delta}\right)$, then with probability $1 - e^{-\Omega\left(k \log \frac{Lr}{\delta}\right)}$, it holds that*

$$\left\langle \frac{1}{m}\mathbf{A}^T(\bar{\mathbf{y}} - \mu\mathbf{A}\bar{\mathbf{x}}), \tilde{\mathbf{x}} - \mu\bar{\mathbf{x}} \right\rangle \leq O\left(\psi\sqrt{\frac{k \log \frac{Lr}{\delta}}{m}}\right) \|\tilde{\mathbf{x}} - \mu\bar{\mathbf{x}}\|_2 + O\left(\delta\psi\sqrt{\frac{k \log \frac{Lr}{\delta}}{m}}\right). \tag{9}$$

With the above auxiliary results in place, the proof of Theorem 1 is given as follows.

*Proof of Theorem 1.* Because $\hat{\mathbf{x}}$ is a solution to the $\mathcal{K}$-Lasso and $\mu\mathbf{x}^* \in \mathcal{K}$, we have

$$\|\tilde{\mathbf{y}} - \mu\mathbf{A}\mathbf{x}^*\|_2^2 \geq \|\tilde{\mathbf{y}} - \mathbf{A}\hat{\mathbf{x}}\|_2^2 = \|(\tilde{\mathbf{y}} - \mu\mathbf{A}\mathbf{x}^*) - \mathbf{A}(\hat{\mathbf{x}} - \mu\mathbf{x}^*)\|_2^2, \tag{10}$$

and expanding the square and diving by $m$ gives

$$\frac{1}{m}\|\mathbf{A}(\hat{\mathbf{x}} - \mu\mathbf{x}^*)\|_2^2 \leq \frac{2}{m}\langle \tilde{\mathbf{y}} - \mu\mathbf{A}\mathbf{x}^*, \mathbf{A}(\hat{\mathbf{x}} - \mu\mathbf{x}^*)\rangle \tag{11}$$

$$= \frac{2}{m}\langle \mathbf{A}^T(\tilde{\mathbf{y}} - \mu\mathbf{A}\mathbf{x}^*), \hat{\mathbf{x}} - \mu\mathbf{x}^*\rangle. \tag{12}$$

We aim to derive a suitable lower bound on $\frac{1}{m}\|\mathbf{A}(\hat{\mathbf{x}} - \mu\mathbf{x}^*)\|_2^2$ and an upper bound on $\frac{2}{m}\langle\mathbf{A}^T(\tilde{\mathbf{y}} - \mu\mathbf{A}\mathbf{x}^*), \hat{\mathbf{x}} - \mu\mathbf{x}^*\rangle$. For any $\delta \in (0,1)$ satisfying $Lr = \Omega(\delta n)$ (to be verified later), setting $\alpha = \frac{1}{2}$ in Lemma 1, we have that if $m = \Omega\left(k\log\frac{Lr}{\delta}\right)$, then with probability $1 - e^{-\Omega(m)}$,

$$\left\|\frac{1}{\sqrt{m}}\mathbf{A}(\hat{\mathbf{x}} - \mu\mathbf{x}^*)\right\|_2 \geq \frac{1}{2}\|\hat{\mathbf{x}} - \mu\mathbf{x}^*\|_2 - \delta. \tag{13}$$

Taking the square on both sides and re-arranging, we obtain

$$\frac{1}{4}\|\hat{\mathbf{x}} - \mu\mathbf{x}^*\|_2^2 \leq \frac{1}{m}\|\mathbf{A}(\hat{\mathbf{x}} - \mu\mathbf{x}^*)\|_2^2 + \delta\|\hat{\mathbf{x}} - \mu\mathbf{x}^*\|_2. \tag{14}$$

Recalling that $\mathbf{y} = f(\mathbf{A}\mathbf{x}^*)$, we also have

$$\frac{1}{m}\langle\mathbf{A}^T(\tilde{\mathbf{y}} - \mu\mathbf{A}\mathbf{x}^*), \hat{\mathbf{x}} - \mu\mathbf{x}^*\rangle = \frac{1}{m}\langle\mathbf{A}^T(\tilde{\mathbf{y}} - \mathbf{y}), \hat{\mathbf{x}} - \mu\mathbf{x}^*\rangle + \frac{1}{m}\langle\mathbf{A}^T(\mathbf{y} - \mu\mathbf{A}\mathbf{x}^*), \hat{\mathbf{x}} - \mu\mathbf{x}^*\rangle. \tag{15}$$

To bound the first term, note that using $\alpha = \frac{1}{2}$ in Lemma 2, we have with probability $1 - e^{-\Omega(m)}$ that

$$\left\|\frac{1}{\sqrt{m}}\mathbf{A}(\hat{\mathbf{x}} - \mu\mathbf{x}^*)\right\|_2 \leq O(\|\hat{\mathbf{x}} - \mu\mathbf{x}^*\|_2 + \delta). \tag{16}$$

Therefore, the following holds with probability $1 - e^{-\Omega(m)}$:

$$\left\langle\frac{1}{m}\mathbf{A}^T(\mathbf{y} - \tilde{\mathbf{y}}), \hat{\mathbf{x}} - \mu\mathbf{x}^*\right\rangle = \left\langle\frac{1}{\sqrt{m}}(\mathbf{y} - \tilde{\mathbf{y}}), \frac{1}{\sqrt{m}}\mathbf{A}(\hat{\mathbf{x}} - \mu\mathbf{x}^*)\right\rangle \tag{17}$$

$$\leq \left\|\frac{1}{\sqrt{m}}(\mathbf{y} - \tilde{\mathbf{y}})\right\|_2 \cdot \left\|\frac{1}{\sqrt{m}}\mathbf{A}(\hat{\mathbf{x}} - \mu\mathbf{x}^*)\right\|_2 \tag{18}$$

$$\leq \tau O(\|\hat{\mathbf{x}} - \mu\mathbf{x}^*\|_2 + \delta) \tag{19}$$

by (16) and the assumption $\frac{1}{\sqrt{m}}\|\tilde{\mathbf{y}} - \mathbf{y}\|_2 \leq \tau$.

We now consider the second term in (15). From Lemma 3, we have that when $Lr = \Omega(\delta n)$ and $m = \Omega\left(k\log\frac{Lr}{\delta}\right)$, with probability $1 - e^{-\Omega\left(k\log\frac{Lr}{\delta}\right)}$,

$$\left\langle\frac{1}{m}\mathbf{A}^T(\mathbf{y} - \mu\mathbf{A}\mathbf{x}^*), \hat{\mathbf{x}} - \mu\mathbf{x}^*\right\rangle \leq O\left(\psi\sqrt{\frac{k\log\frac{Lr}{\delta}}{m}}\right)\|\hat{\mathbf{x}} - \mu\mathbf{x}^*\|_2 + O\left(\psi\delta\sqrt{\frac{k\log\frac{Lr}{\delta}}{m}}\right). \tag{20}$$

Putting the preceding findings together, we have the following with probability $1 - e^{-\Omega\left(k\log\frac{Lr}{\delta}\right)}$:

$$\|\mu\mathbf{x}^* - \hat{\mathbf{x}}\|_2^2 \leq \frac{4}{m}\|\mathbf{A}(\hat{\mathbf{x}} - \mu\mathbf{x}^*)\|_2^2 + 4\delta\|\hat{\mathbf{x}} - \mu\mathbf{x}^*\|_2 \tag{21}$$

$$\leq 8\left\langle\frac{1}{m}\mathbf{A}^T(\tilde{\mathbf{y}} - \mu\mathbf{A}\mathbf{x}^*), \hat{\mathbf{x}} - \mu\mathbf{x}^*\right\rangle + 4\delta\|\hat{\mathbf{x}} - \mu\mathbf{x}^*\|_2 \tag{22}$$

$$\leq O\left(\psi\sqrt{\frac{k\log\frac{Lr}{\delta}}{m}} + \delta + \tau\right)\|\hat{\mathbf{x}} - \mu\mathbf{x}^*\|_2 + O\left(\tau\delta + \psi\delta\sqrt{\frac{k\log\frac{Lr}{\delta}}{m}}\right), \tag{23}$$

where (21) uses (14), (22) uses (12), and (23) combines (15), (19) and (20).

By considering both possible cases of which of the two terms in (23) is larger, we find that if

$$\delta = O\left(\psi\sqrt{\frac{k\log\frac{Lr}{\delta}}{m}}\right), \tag{24}$$

then we have

$$\|\mu\mathbf{x}^* - \hat{\mathbf{x}}\|_2 \leq O\left(\psi\sqrt{\frac{k\log\frac{Lr}{\delta}}{m}} + \tau\right). \tag{25}$$

Then, for any $\epsilon \in (0,1)$ satisfying $Lr = \Omega(\epsilon\psi n)$ (as assumed in the theorem), setting $m = \Omega\left(\frac{k}{\epsilon^2}\log\frac{Lr}{\delta}\right)$ leads to $\delta = O\left(\psi\sqrt{\frac{k\log\frac{Lr}{\delta}}{m}}\right) = O(\epsilon\psi)$ and thus $Lr = \Omega(\delta n)$ (as assumed previously in the proof). Hence, we have with probability $1 - e^{-\Omega(\epsilon^2 m)}$ that

$$\|\mu\mathbf{x}^* - \hat{\mathbf{x}}\|_2 \le \epsilon\psi + \tau. \tag{26}$$

$\square$

## 3.2 Application to Neural Network Generative Models

In the following, we apply Theorem 1 to neural network models, as these are of particular practical interest. We consider feedforward neural network generative models; with $d$ layers, we have

$$G(\mathbf{z}) = \phi_d\left(\phi_{d-1}\left(\cdots\phi_2(\phi_1(\mathbf{z},\boldsymbol{\theta}_1),\boldsymbol{\theta}_2)\cdots,\boldsymbol{\theta}_{d-1}\right),\boldsymbol{\theta}_d\right), \tag{27}$$

where $\mathbf{z} \in B_2^k(r)$, $\phi_i(\cdot)$ is the functional mapping corresponding to the $i$-th layer, and $\boldsymbol{\theta}_i = (\mathbf{W}_i, \mathbf{b}_i)$ is the parameter pair for the $i$-th layer: $\mathbf{W}_i \in \mathbb{R}^{n_i \times n_{i-1}}$ is the matrix of weights, and $\mathbf{b}_i \in \mathbb{R}^{n_i}$ is the vector of offsets, where $n_i$ is the number of neurons in the $i$-th layer. Note that $n_0 = k$ and $n_d = n$. Defining $\mathbf{z}^0 = \mathbf{z}$ and $\mathbf{z}^i = \phi_i(\mathbf{z}^{i-1}, \boldsymbol{\theta}_i)$, we set $\phi_i(\mathbf{z}^{i-1}, \boldsymbol{\theta}_i) = \phi_i(\mathbf{W}_i\mathbf{z}^{i-1} + \mathbf{b}_i)$, $i = 1, 2, \ldots, d$, for some activation function $\phi_i(\cdot)$ applied element-wise.

The following corollary applies Theorem 1 to feedforward neural network generative models. Note that here we do not constrain the $\ell_2$-norm of the signal $G(\mathbf{z}^*) \in \mathcal{K}$.

**Corollary 1.** *Suppose that the generative model $G : B_2^k(r) \to \mathbb{R}^n$ is defined as in (27) with at most $w$ nodes per layer. Suppose that all weights are upper bounded by $W_{\max}$ in absolute value, and that the activation function is 1-Lipschitz. For any $\mathbf{z}^* \in B_2^k(r)$, let $\mathbf{y} = f\left(\mathbf{A}\frac{G(\mathbf{z}^*)}{\mu}\right)$ and let $\tilde{\mathbf{y}}$ be the observed vector with $\frac{1}{\sqrt{m}}\|\mathbf{y} - \tilde{\mathbf{y}}\|_2 \le \tau$. In addition, define $\bar{f}(x) = f\left(\frac{\|G(\mathbf{z}^*)\|_2}{\mu}x\right)$ and $\bar{\mu} = \mathbb{E}[\bar{f}(g)g]$, $\bar{\psi} = \|\bar{f}(g)\|_{\psi_2}$. Then, for any $\epsilon > 0$, if $(wW_{\max})^d r = \Omega(\epsilon\bar{\psi}n)$, $m = \Omega\left(\frac{k}{\epsilon^2}\log\frac{r(wW_{\max})^d}{\epsilon\bar{\psi}}\right)$ and $\frac{\bar{\mu}G(\mathbf{z}^*)}{\|G(\mathbf{z}^*)\|_2} \in \mathcal{K}$, then with probability $1 - e^{-\Omega(\epsilon^2 m)}$, any solution $\hat{\mathbf{x}}$ to the $\mathcal{K}$-Lasso (5) satisfies*

$$\left\|\bar{\mu}\frac{G(\mathbf{z}^*)}{\|G(\mathbf{z}^*)\|_2} - \hat{\mathbf{x}}\right\|_2 \le \epsilon\bar{\psi} + \tau. \tag{28}$$

*Proof.* We know that under the assumptions of the corollary, the generative model $G$ is $L$-Lipschitz with $L = (wW_{\max})^d$ (*cf.* [2, Lemma 8.5]). Letting $\rho = \|G(\mathbf{z}^*)\|_2$, it is straightforward to see that $\bar{f}(g) = f\left(\frac{\rho}{\mu}g\right)$ is also sub-Gaussian, where $g \sim \mathcal{N}(0,1)$. In addition, we have

$$\mathbf{y} = f\left(\mathbf{A}\frac{G(\mathbf{z}^*)}{\mu}\right) = f\left(\mathbf{A}\frac{\rho}{\mu}\cdot\frac{G(\mathbf{z}^*)}{\rho}\right) = \bar{f}\left(\mathbf{A}\frac{G(\mathbf{z}^*)}{\rho}\right). \tag{29}$$

Note that $\frac{G(\mathbf{z}^*)}{\rho}$ is a unit vector, and $\bar{\mu}\frac{G(\mathbf{z}^*)}{\rho} \in \mathcal{K}$ by assumption. Applying Theorem 1 to the observation function $\bar{f}$ and the unit signal vector $\frac{G(\mathbf{z}^*)}{\rho}$ completes the proof. $\square$

Several commonly-used activation functions are 1-Lipschitz, such as i) the ReLU function, $\phi_i(x) = \max(x, 0)$; (ii) the Sigmoid function, $\phi_i(x) = \frac{1}{1+e^{-x}}$; and (iii) the Hyperbolic tangent function with $\phi_i(x) = \frac{e^x - e^{-x}}{e^x + e^{-x}}$. Moreover, it is straightforward to generalize to other activation functions whose Lipschitz constants may exceed one.

The assumptions in Corollary 1 pose some limitations, but are satisfied in several cases of interest. For example, the assumption $\frac{\bar{\mu}G(\mathbf{z}^*)}{\|G(\mathbf{z}^*)\|_2} \in \mathcal{K}$ is satisfied when the generative model is a ReLU network with no offsets (see [39, Remark 2.1]), due to $\mathcal{K}$ being cone-shaped. In addition, while the sub-Gaussianity constant $\bar{\psi} = \|\bar{f}(g)\|_{\psi_2}$ is dependent on $\mathbf{z}^*$, it can be upper bounded independently of $\mathbf{z}^*$ under any observation model in which the measurements are uniformly bounded (e.g., including not only 1-bit, but also more general multi-bit quantized models).

# 4 Variations and Extensions

In this section, we discuss several variations and extensions of our main result, including considering bounded $k$-sparse vectors in Section 4.1, an unknown covariance matrix for the random measurement vectors in Section 4.2, relaxing the norm restriction for the underlying signal in Section 4.3. Some additional variations are given in the appendices, namely, guarantees for a distinct correlation-based optimization algorithm under binary observations (Appendix D), and connections between our sample complexity and the Gaussian mean width (Appendix E).

## 4.1 Bounded Sparse Vectors

In the proof of Theorem 1, for the set of signals $\mathcal{K} = G(B_2^k(r))$, we make use of the property that for any $\delta > 0$, there exists a $\delta$-net $\mathcal{M}$ of $\mathcal{K}$ such that $|\mathcal{M}| \leq O\left(\exp\left(k \log \frac{Lr}{\delta}\right)\right)$. Hence, we can readily extend the result to other sets $\mathcal{K}$ with known bounds on the size of a $\delta$-net. As an example, we state the following for bounded sparse vectors, defining $\Sigma_k^n$ to be the set of $k$-sparse vectors in $\mathbb{R}^n$. A proof outline is given in Appendix C.1.

**Corollary 2.** *Fix $\epsilon > 0$, and let $\nu \geq \mu$ satisfy $\nu = \Omega(\epsilon \psi k)$. Fix $\mathbf{x}^* \in \Sigma_k^n \cap \mathcal{S}^n$, let $\mathbf{y} = f(\mathbf{A}\mathbf{x}^*)$, and let $\tilde{\mathbf{y}}$ be a vector satisfying $\frac{1}{\sqrt{m}}\|\mathbf{y} - \tilde{\mathbf{y}}\|_2 \leq \tau$. Then, when $m = \Omega\left(\frac{k}{\epsilon^2} \log \frac{\nu n}{\epsilon \psi k}\right)$, with probability $1 - e^{-\Omega(\epsilon^2 m)}$, any $\hat{\mathbf{x}}$ that minimizes $\|\tilde{\mathbf{y}} - \mathbf{A}\mathbf{x}\|_2$ over $\Sigma_k^n \cap \nu B_2^n$ satisfies*

$$\|\mu\mathbf{x}^* - \hat{\mathbf{x}}\|_2 \leq \psi\epsilon + \tau. \tag{30}$$

This corollary is similar to other sparsity based results for the generalized Lasso, such as those in [25, 26]. It is intuitive that similar sparsity-based results to Theorem 1 follow without difficulty, given that generative models are known that can produce bounded sparse signals [15, 18].

## 4.2 General Covariance Matrices

Thus far, we have focused on the case that $\mathbf{a}_i \sim \mathcal{N}(\mathbf{0}, \mathbf{I})$. Following the ideas of [25], we can also consider the more general scenario in which $\mathbf{a}_i \sim \mathcal{N}(\mathbf{0}, \boldsymbol{\Sigma})$ for an *unknown* covariance matrix $\boldsymbol{\Sigma} \in \mathbb{R}^{n \times n}$, assuming that $\|\sqrt{\boldsymbol{\Sigma}}\mathbf{x}^*\|_2 = 1$ and $\mu\mathbf{x}^* \in \mathcal{K}$. The definitions of $\mu$ and $\psi$ remain the same, *cf.*, Section 2.1. The following is easily deduced from Theorem 1; see Appendix C.2 for the details.

**Corollary 3.** *Suppose that $\mathbf{a}_i \stackrel{i.i.d.}{\sim} \mathcal{N}(\mathbf{0}, \boldsymbol{\Sigma})$ for $i \in [m]$ and $\|\sqrt{\boldsymbol{\Sigma}}\mathbf{x}^*\|_2 = 1$. Suppose that $\mathbf{y} = f(\mathbf{A}\mathbf{x}^*)$ and $\mu\mathbf{x}^* \in \mathcal{K}$. Let $\tilde{\mathbf{y}}$ be any vector of corrupted measurements satisfying $\frac{1}{\sqrt{m}}\|\mathbf{y} - \tilde{\mathbf{y}}\|_2 \leq \tau$. Then, for any $\epsilon > 0$, when $\|\boldsymbol{\Sigma}\|_{2 \to 2}^{1/2} Lr = \Omega(\epsilon \psi n)$ and $m = \Omega\left(\frac{k}{\epsilon^2} \log \frac{\|\boldsymbol{\Sigma}\|_{2 \to 2}^{1/2} Lr}{\epsilon \psi}\right)$, with probability $1 - e^{-\Omega(\epsilon^2 m)}$, any solution to the generalized Lasso (5) satisfies*

$$\|\sqrt{\boldsymbol{\Sigma}}(\hat{\mathbf{x}} - \mu\mathbf{x}^*)\|_2 \leq \psi\epsilon + \tau. \tag{31}$$

## 4.3 Removing the $\ell_2$-norm Assumption

Continuing from the previous subsection and again following [25], our results can easily be generalized to the case that $\|\sqrt{\boldsymbol{\Sigma}}\mathbf{x}^*\|_2 \neq 1$ (or for $\boldsymbol{\Sigma} = \mathbf{I}$, the case that $\|\mathbf{x}^*\|_2 \neq 1$). The idea is similar to that presented in the proof of Corollary 1. In particular, setting $\rho = \|\sqrt{\boldsymbol{\Sigma}}\mathbf{x}^*\|_2$ and $\bar{\mathbf{x}} = \frac{\mathbf{x}^*}{\rho}$ gives

$$f(\mathbf{A}\mathbf{x}^*) = f(\rho\mathbf{A}\bar{\mathbf{x}}) = \bar{f}(\mathbf{A}\bar{\mathbf{x}}), \tag{32}$$

where $\bar{f}(x) := f(\rho x)$ for $x \in \mathbb{R}$. Hence, for $g \sim \mathcal{N}(0, 1)$, if $\mathbb{E}[\bar{f}(g)g]\bar{\mathbf{x}} \in \mathcal{K}$, the preceding theorems and corollaries apply to the estimation of $\bar{\mathbf{x}}$, with modified parameters

$$\bar{\mu} := \mathbb{E}[\bar{f}(g)g], \quad \bar{\psi} := \|\bar{f}(g)\|_{\psi_2}. \tag{33}$$

# 5 Uniform Recovery Guarantees

In this section, we turn to uniform recovery guarantees, stating that a single matrix $\mathbf{A}$ simultaneously permits the recovery of all $\mathbf{x}^*$ in the set of interest. For brevity, we consider $\mu$ and $\psi$ to be fixed constants and omit them in the $O(\cdot)$ notation.

Our result will depend on the following Local Embedding Property (LEP).

**Definition 3.** *A deterministic function $\tilde{f}$ and measurement matrix $\tilde{\mathbf{A}}$ are said to satisfy the LEP$(S, \delta, \beta)$ with set $S$ and parameters $\delta \geq 0$ and $\beta \geq 0$ if, for any $\mathbf{x}_1 \in S$ and $\mathbf{x}_2 \in S$ satisfying $\|\mathbf{x}_1 - \mathbf{x}_2\|_2 \leq \delta$, the following holds:*

$$\frac{1}{\sqrt{m}}\|\tilde{f}(\tilde{\mathbf{A}}\mathbf{x}_1) - \tilde{f}(\tilde{\mathbf{A}}\mathbf{x}_2)\|_2 \leq C\delta^\beta \tag{34}$$

*for some $C > 0$ not depending on $\delta$.*

This definition essentially states that nearby signals remain close upon multiplying by $\tilde{\mathbf{A}}$ and then applying the function $\tilde{f}$. See, for example, [17, 19] for similar concepts in earlier works. With this definition in place, our main assumption in this section is stated as follows.

**Assumption 1.** *Under the (possibly random) function $f$, i.i.d. Gaussian measurement matrix $\mathbf{A}$, and generative model $G$ with $\mathcal{K} = \mathrm{Range}(G)$, there exists a constant $\beta \in (0, 1]$ and functions $M_{\mathrm{LEP}}, P_{\mathrm{LEP}}$ such that for any sufficiently small $\delta$, the following holds with probability $1 - P_{\mathrm{LEP}}(\delta, \beta)$ when $m \geq M_{\mathrm{LEP}}(\delta, \beta)$: The pair $(f, \mathbf{A})$ satisfies the LEP$(S, \delta, \beta)$ with $S = \mathcal{S}^{n-1} \cap \{\mathbf{x} : c\mathbf{x} \in \mathcal{K} \text{ for some } c \in [\mu(1 - \eta), \mu(1 + \eta)]\}$, where $\eta > 0$ is a (small) positive constant not depending on $\delta$, and $\mu = \mathbb{E}[f(g)g]$.*

While Assumption 1 is somewhat technical, the intuition behind it is simply that if $\mathbf{x}_1$ is close to $\mathbf{x}_2$, then $f(\mathbf{A}\mathbf{x}_1)$ is close to $f(\mathbf{A}\mathbf{x}_2)$. We restrict $\beta \leq 1$ because the case $\beta > 1$ fails even for linear measurements, and the LEP for $\beta > 1$ implies the same for $\beta = 1$. Before providing some examples of models satisfying Assumption 1, we state our uniform recovery result, proved in Appendix G.

**Theorem 2.** *Suppose that $f$ yields parameters $\mu = \Theta(1)$ and $\psi = \Theta(1)$, and that Assumption 1 holds. Then, for sufficiently small $\epsilon > 0$, if $Lr = \Omega(\epsilon n)$ and $m \geq M_{\mathrm{LEP}}\left(\mathcal{K}, \epsilon^{1/\beta}, \beta\right) + \Omega\left(\frac{k}{\epsilon^2}\log\frac{Lr}{\epsilon}\right)$, then with probability $1 - e^{-\Omega(m)} - P_{\mathrm{LEP}}\left(\mathcal{K}, \epsilon^{1/\beta}, \beta\right)$, we have the following: For any signal $\mathbf{x}^* \in \mathcal{S}^{n-1}$ with $\mu\mathbf{x}^* \in \mathcal{K}$ and $\mathbf{y} = f(\mathbf{A}\mathbf{x}^*)$, and any vector $\tilde{\mathbf{y}}$ of corrupted measurement satisfying $\frac{1}{\sqrt{m}}\|\tilde{\mathbf{y}} - \mathbf{y}\|_2 \leq \tau$, any solution $\hat{\mathbf{x}}$ to the $\mathcal{K}$-Lasso satisfies*

$$\|\mu\mathbf{x}^* - \hat{\mathbf{x}}\|_2 \leq \epsilon + \tau. \tag{35}$$

Assumption 1 is satisfied by various measurement models; for example:

- Under the linear model $f(x) = x$, setting $\alpha = \frac{1}{2}$ in Lemma 2, choosing $\delta > 0$, and setting $\beta = 1$ and $\mu = 1$, we obtain $M_{\mathrm{LEP}}(\delta, \beta) = O\left(k\log\frac{Lr}{\delta}\right)$ and $P_{\mathrm{LEP}}(\delta, \beta) = e^{-\Omega(m)}$.
- The preceding example directly extends to any 1-Lipschitz function $f$, such as the censored Tobit model with $f(x) = \max\{x, 0\}$.
- In Appendix F, we use an existing result in [17] to show that for the noiseless 1-bit model with $f(x) = \mathrm{sign}(x)$, we can choose any $\delta = O(1)$, set $\beta = \frac{1}{2}$ and $\mu = \sqrt{\frac{2}{\pi}}$, and obtain $M_{\mathrm{LEP}}(\delta, \beta) = O\left(\frac{k}{\delta}\log\frac{Lr}{\delta^2}\right)$ and $P_{\mathrm{LEP}}(\delta, \beta) = e^{-\Omega(\delta m)}$.

Regarding the last of these, we note that our sample complexity in Theorem 2 matches that of [17, Corollary 3]. An advantage of our result compared to [17] is that the $\mathcal{K}$-Lasso objective function can be optimized directly using gradient methods, whereas the Hamming distance based objective proposed in [17] appears to be difficult to use directly in practice. Instead, it is proposed in [17] to first approximate the objective by a convex one, and then apply a sub-gradient based method.

## 6 Conclusion

We have provided recovery guarantees for the generalized Lasso with nonlinear observations and generative priors. In particular, we showed that under i.i.d. Gaussian measurements, roughly $O\left(\frac{k}{\epsilon^2}\log L\right)$ samples suffice for non-uniform $\epsilon$-recovery, with robustness to adversarial noise. Moreover, we derived a uniform recovery guarantee under the assumption of the local embedding property. Possible extensions for future work include handling signals with representation error (i.e., $\mu\mathbf{x}^*$ is not quite in $\mathcal{K}$) [2, 25], a sharp analysis including constants [21, 22, 30, 31], and lower bounds on the sample complexity [15, 18, 26].

**Acknowledgment.** This work was supported by the Singapore National Research Foundation (NRF) under grant number R-252-000-A74-281.

## Broader Impact

**Who may benefit from this research.** This is a theory paper primarily targeted at the research community. The signal recovery techniques studied could potentially be useful for practitioners in areas such as image processing, audio processing, and medical imaging.

**Who may be put at disadvantage from this research.** We are not aware of any significant/imminent risks of placing anyone at a disadvantage.

**Consequences of failure of the system.** We believe that most failures should be immediately evident and detectable due to visibly poor reconstruction performance, and any such outputs could be discarded as needed. However, some more subtle issues could arise, such as the reconstruction missing important details in the signal due to the generative model not capturing them. As a result, care is advised in the choice of generative model, particularly in applications for which the reconstruction of fine details is crucial.

**Potential biases.** The signal recovery algorithm that we consider takes as input an arbitrary pre-trained generative model. If such a pre-trained model has inherent biases, they could be transferred to the signal recovery algorithm.

## Footnotes

[1]A *uniform recovery* guarantee is one in which some measurement matrix $\mathbf{A}$ simultaneously ensures the recovery of all $\mathbf{x}^*$ in the set of interest. In contrast, *non-uniform recovery* only requires a randomly-drawn $\mathbf{A}$ to succeed with high probability for fixed $\mathbf{x}^*$.

[2]A set $\mathcal{K}$ is star-shaped if $\lambda \mathcal{K} \subseteq \mathcal{K}$ whenever $0 \leq \lambda \leq 1$.

[3] Here and in subsequent results, the implied constants in these $\Omega(\cdot)$ terms are implicitly assumed to be sufficiently large. Regarding the assumption $Lr = \Omega(\epsilon\psi n)$, we note that $d$-layer neural networks typically give $L = n^{\Theta(d)}$ [2], meaning this assumption is certainly satisfied for fixed $(r, \psi, \epsilon)$.

[4]For example, we may choose $\tilde{\mathbf{x}}$ to be a minimizer of the $\mathcal{K}$-Lasso (5) with inputs $\bar{\mathbf{y}}$ and $\mathbf{A}$.

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
