[Supplementary Material]

# Supplementary Material

## The Generalized Lasso with Nonlinear Observations
## and Generative Priors (NeurIPS 2020)

### Zhaoqiang Liu and Jonathan Scarlett

## A    Table Comparing to the Existing Literature

The comparison of our results to those in the existing literature, as discussed in Section 1.1, is outlined in Table 1. In the table, we write $\mu = \mathbb{E}[f(g)g]$ for $g \sim \mathcal{N}(0,1)$. We use $\mathcal{K}$ to represent the structured set of interest, and $\Sigma_k^n$ to represent the set of $k$-sparse vectors in $\mathbb{R}^n$. For Projected Back Projection (PBP) [26], the reconstructed vector is $\hat{\mathbf{x}} := \mathcal{P}_{\mathcal{K}}\left(\frac{1}{m}\mathbf{A}^T\mathbf{y}\right)$, where $\mathcal{P}_{\mathcal{K}}$ is the projection operator onto $\mathcal{K}$. In addition, $\partial\mathcal{K}$ represents the boundary of $\mathcal{K}$. Letting $q : \mathbb{R}^n \to \mathbb{R}$ be the density of the random measurement vector $\mathbf{a}$ and assume that $q$ is differentiable, we write $S_q(\mathbf{a}) = -\frac{\nabla q(\mathbf{a})}{q(\mathbf{a})}$. For thresholded Empirical Risk Minimization (ERM), the reconstructed vector is $\hat{\mathbf{x}} := \arg\min_{\mathbf{x} \in G(B_2^k(r))} \|\mathbf{x}\|_2^2 - \frac{2}{m}\sum_{i=1}^m \hat{y}_i \langle S_q(\mathbf{a}_i), \mathbf{x} \rangle$, where $\hat{y}_i := \text{sign}(y_i) \cdot |y_i| \wedge \tau$ for some thresholding parameter $\tau$. We recall that GMW stands for Gaussian Mean Width (*cf.*, Appendix E) and LEP stands for Local Embedding Property (*cf.*, Definition 3). Interested readers may refer to [40, Table 1] for a summary of further relevant results.

## B    Omitted Details and Additional Auxiliary Results for Proving Theorem 1 (Non-Uniform Recovery)

In this section, we fill in the missing details for proving Theorem 1, including a statement of the concentration bound used to establish Lemma 2, and a proof for Lemma 3. We first provide some useful additional auxiliary results that are general, and then some that are specific to our setup.

### B.1    General Auxiliary Results

We have the following basic concentration inequality, which is used in the proof of Lemma 2.

**Lemma 4.** ([34, Lemma 1.3]) *Fix fixed* $\mathbf{x} \in \mathbb{R}^n$, *we have for any* $\epsilon \in (0,1)$ *that*

$$\mathbb{P}\left((1-\epsilon)\|\mathbf{x}\|_2^2 \leq \left\|\frac{1}{\sqrt{m}}\mathbf{A}\mathbf{x}\right\|_2^2 \leq (1+\epsilon)\|\mathbf{x}\|_2^2\right) \geq 1 - 2e^{-\epsilon^2(1-\epsilon)m/4}. \tag{36}$$

The following definition formally introduces the notion of an $\epsilon$-net, also known as a covering set.

**Definition 4.** *Let* $(\mathcal{X}, d)$ *be a metric space, and fix* $\epsilon > 0$. *A subset* $S \subseteq \mathcal{X}$ *is said be an* $\epsilon$-net *of* $\mathcal{X}$ *if, for all* $\mathbf{x} \in \mathcal{X}$, *there exists some* $\mathbf{s} \in S$ *such that* $d(\mathbf{x}, \mathbf{s}) \leq \epsilon$. *The minimal cardinality of an* $\epsilon$-net *of* $\mathcal{X}$ *is denoted by* $\mathcal{N}^*(\mathcal{X}, \epsilon)$ *and is called the* covering number *of* $\mathcal{X}$ *(with parameter* $\epsilon$).

Alongside the sub-Gaussian notion in Definition 1, we use the following definition of a sub-exponential random variable and sub-exponential norm.

**Definition 5.** *A random variable* $X$ *is said to be sub-exponential if there exists a positive constant* $C$ *such that* $(\mathbb{E}[|X|^p])^{\frac{1}{p}} \leq Cp$ *for all* $p \geq 1$. *The sub-exponential norm of* $X$ *is defined as*

$$\|X\|_{\psi_1} = \sup_{p \geq 1} p^{-1}\left(\mathbb{E}[|X|^p]\right)^{\frac{1}{p}}. \tag{37}$$

The product of two sub-Gaussian random variables is sub-exponential, as stated in the following.

**Lemma 5.** ([36, Lemma 2.7.7]) *Let* $X$ *and* $Y$ *be sub-Gaussian random variables (not necessarily independent). Then* $XY$ *is sub-exponential, and satisfies*

$$\|XY\|_{\psi_1} \leq \|X\|_{\psi_2}\|Y\|_{\psi_2}. \tag{38}$$

Table 1: Summary of existing results and their associated conditions on the structured set, the observed signal, the sensing model, and the reconstruction algorithm.

| | [25] | [26] | [10] | [39] | (this work) |
|---|---|---|---|---|---|
| Signal set $\mathcal{K} \subseteq \mathbb{R}^n$ | Convex; considers (local) GMW on the tangent cone at $\mathbf{x}^*$ | Star-shaped, closed; considers (local) GMW | Convex; considers (local) GMW | $G(B_2^k(r))$ | $G(B_2^k(r))$ |
| Condition on $\mathbf{x}^*$ | $\mathbf{x}^* \in \mathcal{S}^{n-1} \cap \frac{1}{\mu}\mathcal{K}$ | $\mu\frac{\mathbf{x}^*}{\|\mathbf{x}^*\|_2} \in \mathcal{K}$ | $\mathbf{x}^* \in \mathcal{S}^{n-1} \cap \frac{1}{\mu}\mathcal{K}$ | $\mathbf{x}^* \in \mathcal{S}^{n-1} \cap \mathcal{K}$, $\lambda\mathbf{x}^* \in \mathcal{K}$, $\lambda := \mathbb{E}\left[f'\left(\mathbf{a}^T\mathbf{x}^*, \xi\right)\right]$ | $\mathbf{x}^* \in \mathcal{S}^{n-1} \cap \frac{1}{\mu}\mathcal{K}$ |
| Sensing model | $f(\mathbf{a}_i^T\mathbf{x}^*)$ | $f(\mathbf{a}_i^T\mathbf{x}^*)$, $y_i$ sub-Gaussian | $f(\mathbf{a}_i^T\mathbf{x}^*)$, with extra adversarial noise | $f(\mathbf{a}_i^T\mathbf{x}^*, \xi_i)$, $\xi_i$ random noise, $f$ differentiable and deterministic | $f(\mathbf{a}_i^T\mathbf{x}^*)$ $y_i$ sub-Gaussian, with extra adversarial noise |
| Algorithm | $\mathcal{K}$-Lasso | PBP | General convex loss functions | Thresholded ERM | $\mathcal{K}$-Lasso |
| Uniform/Non-uniform guarantee | Non-uniform | Non-uniform | Non-uniform | Non-uniform | Non-uniform (and uniform if LEP holds) |
| Error bound | $\mu\mathbf{x}^* \in \partial\mathcal{K}$: the dependence on $m$ can be $m^{-\frac{1}{2}}$ (In general, $m^{-\frac{1}{4}}$) | $\mathcal{K} = \Sigma_k^n$, noiseless 1-bit observations: $\sqrt{\frac{k\log\frac{n}{k}}{m}}$ (In general, $m^{-\frac{1}{4}}$) | $\mathcal{K} = \mu\left(\sqrt{k}B_1^n \cap B_2^n\right)$ and $\mu\mathbf{x}^* \in \partial\mathcal{K}$: $\sqrt{\frac{k\log\frac{n}{k}}{m}}$ (In general, $m^{-\frac{1}{4}}$) | $\sqrt{\frac{k\log Lr}{m}}$ | $\sqrt{\frac{k\log Lr}{m}}$ |
| Measurement vector $\mathbf{a}_i$ | $\mathcal{N}(\mathbf{0}, \mathbf{\Sigma})$ | $\mathcal{N}(\mathbf{0}, \mathbf{I}_n)$ | $\mathcal{N}(\mathbf{0}, \mathbf{\Sigma})$ | $S_q(\mathbf{a})$ sub-Gaussian | $\mathcal{N}(\mathbf{0}, \mathbf{\Sigma})$ |

In our setting, since we assume that $y_i$ is sub-Gaussian and $\langle \mathbf{a}_i, \mathbf{x}^* \rangle \sim \mathcal{N}(0,1)$, Lemma 5 reveals that the random variable $y_i \langle \mathbf{a}_i, \mathbf{x}^* \rangle$ is sub-exponential, and has the same distribution as $f(g)g$ with $g \sim \mathcal{N}(0,1)$, yielding

$$\mu = \mathbb{E}[f(g)g] \le \mathbb{E}[|f(g)g|] \le \|f(g)g\|_{\psi_1} \le C\psi \tag{39}$$

for some absolute constant $C > 0$. In addition, we have the following concentration inequality for sums of independent sub-exponential random variables.

**Lemma 6.** ([35, Proposition 5.16]) *Let* $X_1, \dots, X_N$ *be independent centered sub-exponential random variables, and* $K = \max_i \|X_i\|_{\psi_1}$. *Then for every* $\boldsymbol{\alpha} = [\alpha_1, \dots, \alpha_N]^T \in \mathbb{R}^N$ *and* $\epsilon \ge 0$, *it holds that*

$$\mathbb{P}\left( \left| \sum_{i=1}^N \alpha_i X_i \right| \ge \epsilon \right) \le 2 \exp\left( -c \cdot \min\left( \frac{\epsilon^2}{K^2 \|\boldsymbol{\alpha}\|_2^2}, \frac{\epsilon}{K \|\boldsymbol{\alpha}\|_\infty} \right) \right). \tag{40}$$

### B.2 Auxiliary Results for Our Setup

In the remainder of this appendix, we consider the setup described in Section 2. Based on Lemma 6, we have the following.

**Lemma 7.** *Fix any* $\bar{\mathbf{x}} \in \mathcal{S}^{n-1}$ *and let* $\bar{\mathbf{y}} := f(\mathbf{A}\bar{\mathbf{x}})$. *For any* $t > 0$, *if* $m = \Omega(t + \log n)$, *then with probability* $1 - e^{-\Omega(t)}$, *we have*

$$\left\| \frac{1}{m} \mathbf{A}^T (\bar{\mathbf{y}} - \mu \mathbf{A}\bar{\mathbf{x}}) \right\|_\infty \le O\left( \psi \sqrt{\frac{t + \log n}{m}} \right). \tag{41}$$

*Proof.* For any fixed $j \in [n]$, let $X_j$ be the $j$-th entry of $\frac{1}{m} \mathbf{A}^T (\bar{\mathbf{y}} - \mu \mathbf{A}\bar{\mathbf{x}})$. We have

$$X_j = \frac{1}{m} \sum_{i=1}^m a_{ij} (\bar{y}_i - \mu \langle \mathbf{a}_i, \bar{\mathbf{x}} \rangle) = \frac{1}{m} \sum_{i=1}^m X_{ij}, \tag{42}$$

where $X_{ij} := a_{ij} (\bar{y}_i - \mu \langle \mathbf{a}_i, \bar{\mathbf{x}} \rangle)$. We proceed by showing that $\{X_{ij}\}_{i \in [m]}$ are i.i.d. sub-exponential random variables.

Since $\mathbf{a}_i \sim \mathcal{N}(\mathbf{0}, \mathbf{I}_n)$, we have $\mathrm{Cov}[a_{ij}, \langle \mathbf{a}_i, \bar{\mathbf{x}} \rangle] = \bar{x}_j$. For $i \in [m]$, letting $g := \langle \mathbf{a}_i, \bar{\mathbf{x}} \rangle \sim \mathcal{N}(0,1)$, we find that $a_{ij} \sim \mathcal{N}(0,1)$ can be written as $a_{ij} = \bar{x}_j g + \sqrt{1 - \bar{x}_j^2} h$, where $h \sim \mathcal{N}(0,1)$ is independent of $g$. Thus, $X_{ij} = a_{ij}(\bar{y}_i - \mu \langle \mathbf{a}_i, \bar{\mathbf{x}} \rangle) = (\bar{x}_j g + \sqrt{1 - \bar{x}_j^2} h)(f(g) - \mu g)$, and hence $\mathbb{E}[X_{ij}] = \bar{x}_j \mathbb{E}[f(g)g - \mu g^2] = \mu - \mu = 0$. In addition, from Lemma 5 and (39), we obtain

$$\|X_{ij}\|_{\psi_1} \le C' \|f(g) - \mu g\|_{\psi_2} \le C'' \psi. \tag{43}$$

For fixed $c' > 0$, letting $\epsilon_j = c' \|X_{1j}\|_{\psi_1} \sqrt{\frac{t + \log n}{m}}$ and $\epsilon = \max_j \epsilon_j$, we have from Lemma 6 that

$$\mathbb{P}(|X_j| \ge \epsilon) \le \mathbb{P}(|X_j| \ge \epsilon_j) \tag{44}$$

$$= \mathbb{P}\left( \frac{1}{m} \left| \sum_{i=1}^m X_{ij} \right| \ge \epsilon_j \right) \tag{45}$$

$$\le 2 \exp\left( -c \min\left( \frac{m \epsilon_j^2}{\|X_{1j}\|_{\psi_1}^2}, \frac{m \epsilon_j}{\|X_{1j}\|_{\psi_1}} \right) \right) \tag{46}$$

$$\le \exp\left( -\Omega(t + \log n) \right), \tag{47}$$

where (47) uses $m = \Omega(t + \log n)$ and the choice of $\epsilon_j$. For sufficiently large $c'$, we can make the implied constant to $\Omega(\cdot)$ in (47) greater than one, and taking the union bound over $j \in [n]$ gives

$$\mathbb{P}\left( \left\| \frac{1}{m} \mathbf{A}^T (\bar{\mathbf{y}} - \mu \mathbf{A}\bar{\mathbf{x}}) \right\|_\infty \ge \epsilon \right) \le n \exp\left( -\Omega(t + \log n) \right) = e^{-\Omega(t)} \tag{48}$$

as desired. □

In addition, we have the following useful lemma.

**Lemma 8.** *Fix any $\bar{\mathbf{x}} \in \mathcal{S}^{n-1}$ and let $\bar{\mathbf{y}} := f(\mathbf{A}\bar{\mathbf{x}})$. For any fixed $\mathbf{u} \in \mathbb{R}^n$, the random variable $U := \frac{1}{m} \left\langle \mathbf{u}, \mathbf{A}^T(\bar{\mathbf{y}} - \mu\mathbf{A}\bar{\mathbf{x}}) \right\rangle$ has zero mean and is sub-exponential. Moreover, for any $\xi > 0$, if $m = \Omega(\xi^2)$, then with probability $1 - e^{-\Omega(\xi^2)}$, we have*

$$|U| \leq \frac{\xi\psi\|\mathbf{u}\|_2}{\sqrt{m}}. \tag{49}$$

*Proof.* When $\mathbf{u}$ is the zero vector, the result is trivial, so we only consider $\mathbf{u} \neq \mathbf{0}$. Following similar steps to the proof of Lemma 7, we write

$$\left\langle \mathbf{u}, \mathbf{A}^T(\bar{\mathbf{y}} - \mu\mathbf{A}\bar{\mathbf{x}}) \right\rangle = \sum_{j=1}^{n} u_j \sum_{i=1}^{m} a_{ij}(\bar{y}_i - \mu\langle\mathbf{a}_i, \bar{\mathbf{x}}\rangle) \tag{50}$$

$$= \sum_{i=1}^{m} (\bar{y}_i - \mu\langle\mathbf{a}_i, \bar{\mathbf{x}}\rangle) \sum_{j=1}^{n} u_j a_{ij} \tag{51}$$

$$= \|\mathbf{u}\|_2 \sum_{i=1}^{m} (\bar{y}_i - \mu\langle\mathbf{a}_i, \bar{\mathbf{x}}\rangle)\langle\mathbf{a}_i, \bar{\mathbf{u}}\rangle \tag{52}$$

$$= \|\mathbf{u}\|_2 \sum_{i=1}^{m} U_i, \tag{53}$$

where $\bar{\mathbf{u}} = \frac{\mathbf{u}}{\|\mathbf{u}\|_2}$ and $U_i := (\bar{y}_i - \mu\langle\mathbf{a}_i, \bar{\mathbf{x}}\rangle)\langle\mathbf{a}_i, \bar{\mathbf{u}}\rangle$. We proceed by showing that $U_1, \ldots, U_m$ are i.i.d. sub-exponential random variables. Note that $\langle\mathbf{a}_i, \bar{\mathbf{u}}\rangle \sim \mathcal{N}(0, 1)$, and $\mathrm{Cov}[\langle\mathbf{a}_i, \bar{\mathbf{u}}\rangle, \langle\mathbf{a}_i, \bar{\mathbf{x}}\rangle] = \langle\bar{\mathbf{x}}, \bar{\mathbf{u}}\rangle$. Fixing $i \in [m]$ and letting $g := \langle\mathbf{a}_i, \bar{\mathbf{x}}\rangle \sim \mathcal{N}(0, 1)$, we find that $\langle\mathbf{a}_i, \bar{\mathbf{u}}\rangle$ can be written as $\langle\mathbf{a}_i, \bar{\mathbf{u}}\rangle = \langle\bar{\mathbf{x}}, \bar{\mathbf{u}}\rangle g + \sqrt{1 - \langle\bar{\mathbf{x}}, \bar{\mathbf{u}}\rangle^2}h$, where $h \sim \mathcal{N}(0, 1)$ is independent of $g$. Therefore, we obtain

$$\mathbb{E}[U_i] = \mathbb{E}\left[(\bar{y}_i - \mu\langle\mathbf{a}_i, \bar{\mathbf{x}}\rangle)\langle\mathbf{a}_i, \bar{\mathbf{u}}\rangle\right] = \langle\bar{\mathbf{x}}, \bar{\mathbf{u}}\rangle\mathbb{E}[f(g)g - \mu g^2] = 0. \tag{54}$$

In addition, from Lemma 5 and (39), we derive

$$\|U_i\|_{\psi_1} \leq C'\|f(g) - \mu g\|_{\psi_2} \leq C''\psi. \tag{55}$$

Letting $\epsilon = c'\frac{\xi\psi\|\mathbf{u}\|_2}{\sqrt{m}}$, we deduce from Lemma 6 that

$$\mathbb{P}(|U| \geq \epsilon) = \mathbb{P}\left(\frac{\|\mathbf{u}\|_2}{m}\left|\sum_{i=1}^{m} U_i\right| \geq \epsilon\right) \tag{56}$$

$$\leq 2\exp\left(-c\min\left(\frac{m\epsilon^2}{\|U_i\|_{\psi_1}^2\|\mathbf{u}\|_2^2}, \frac{m\epsilon}{\|U_i\|_{\psi_1}\|\mathbf{u}\|_2}\right)\right) \tag{57}$$

$$\leq e^{-\Omega(\xi^2)}, \tag{58}$$

where (58) follows from $m = \Omega(\xi^2)$ and the choice of $\epsilon$. $\qquad\square$

Based on the above results, we are now in a positive to prove Lemma 3.

### B.3 Proof of Lemma 3 (Main Auxiliary Result for Proving Theorem 1)

We utilize ideas from [2] based on forming a chain of nets. Specifically, for a positive integer $l$, let $M = M_0 \subseteq M_1 \subseteq \ldots \subseteq M_l$ be a chain of nets of $B_2^k(r)$ such that $M_i$ is a $\frac{\delta_i}{L}$-net with $\delta_i = \frac{\delta}{2^i}$. There exists such a chain of nets with [35, Lemma 5.2]

$$\log|M_i| \leq k\log\frac{4Lr}{\delta_i}. \tag{59}$$

By the $L$-Lipschitz assumption on $G$, we have for any $i \in [l]$ that $G(M_i)$ is a $\delta_i$-net of $G(B_2^k(r))$. We write $\tilde{\mathbf{x}}$ as

$$\tilde{\mathbf{x}} = (\tilde{\mathbf{x}} - \tilde{\mathbf{x}}_l) + (\tilde{\mathbf{x}}_l - \tilde{\mathbf{x}}_{l-1}) + \ldots + (\tilde{\mathbf{x}}_1 - \tilde{\mathbf{x}}_0) + \tilde{\mathbf{x}}_0, \tag{60}$$

where $\tilde{\mathbf{x}}_i \in G(M_i)$ for all $i \in [l]$, and $\|\tilde{\mathbf{x}} - \tilde{\mathbf{x}}_l\|_2 \le \frac{\delta}{2^r}$, $\|\tilde{\mathbf{x}}_i - \tilde{\mathbf{x}}_{i-1}\|_2 \le \frac{\delta}{2^{i-1}}$ for all $i \in [l]$. Therefore, the triangle inequality gives

$$\|\tilde{\mathbf{x}} - \tilde{\mathbf{x}}_0\|_2 < 2\delta. \tag{61}$$

We decompose $\frac{1}{m}\langle \mathbf{A}^T(\bar{\mathbf{y}} - \mu\mathbf{A}\bar{\mathbf{x}}), \tilde{\mathbf{x}} - \mu\bar{\mathbf{x}}\rangle$ into three terms:

$$\left\langle \frac{1}{m}\mathbf{A}^T(\bar{\mathbf{y}} - \mu\mathbf{A}\bar{\mathbf{x}}), \tilde{\mathbf{x}} - \mu\bar{\mathbf{x}}\right\rangle = \left\langle \frac{1}{m}\mathbf{A}^T(\bar{\mathbf{y}} - \mu\mathbf{A}\bar{\mathbf{x}}), \tilde{\mathbf{x}}_0 - \mu\bar{\mathbf{x}}\right\rangle$$

$$+ \sum_{i=1}^{l}\left\langle \frac{1}{m}\mathbf{A}^T(\bar{\mathbf{y}} - \mu\mathbf{A}\bar{\mathbf{x}}), \tilde{\mathbf{x}}_i - \tilde{\mathbf{x}}_{i-1}\right\rangle$$

$$+ \left\langle \frac{1}{m}\mathbf{A}^T(\bar{\mathbf{y}} - \mu\mathbf{A}\bar{\mathbf{x}}), \tilde{\mathbf{x}} - \tilde{\mathbf{x}}_l\right\rangle. \tag{62}$$

We derive upper bounds for these terms separately:

1. For any $\mathbf{t} \in \mathbb{R}^n$, from Lemma 8, we have that for any $\xi > 0$, if $m = \Omega(\xi^2)$, then with probability $1 - e^{-\Omega(\xi^2)}$,

$$\left\langle \frac{1}{m}\mathbf{A}^T(\bar{\mathbf{y}} - \mu\mathbf{A}\bar{\mathbf{x}}), \mathbf{t} - \mu\bar{\mathbf{x}}\right\rangle \le \frac{\xi\psi}{\sqrt{m}}\|\mathbf{t} - \mu\bar{\mathbf{x}}\|_2. \tag{63}$$

Recall that $\log|G(M)| = \log|M| \le k\log\frac{4Lr}{\delta}$. We set $\xi = C\sqrt{k\log\frac{Lr}{\delta}}$ in (63), where $C$ is a certain positive constant, and let $m = \Omega(\xi^2) = \Omega(k\log\frac{Lr}{\delta})$. By the union bound over $G(M)$, we have that with probability $1 - e^{-\Omega(k\log\frac{Lr}{\delta})}$, for *all* $\mathbf{t} \in G(M)$,

$$\left\langle \frac{1}{m}\mathbf{A}^T(\bar{\mathbf{y}} - \mu\mathbf{A}\bar{\mathbf{x}}), \mathbf{t} - \mu\bar{\mathbf{x}}\right\rangle \le O\left(\psi\sqrt{\frac{k\log\frac{Lr}{\delta}}{m}}\right)\|\mathbf{t} - \mu\bar{\mathbf{x}}\|_2. \tag{64}$$

Therefore, with probability $1 - e^{-\Omega(k\log\frac{Lr}{\delta})}$, the first term in (62) can be upper bounded by

$$\left\langle \frac{1}{m}\mathbf{A}^T(\bar{\mathbf{y}} - \mu\mathbf{A}\bar{\mathbf{x}}), \tilde{\mathbf{x}}_0 - \mu\bar{\mathbf{x}}\right\rangle \le O\left(\psi\sqrt{\frac{k\log\frac{Lr}{\delta}}{m}}\right)\|\tilde{\mathbf{x}}_0 - \mu\bar{\mathbf{x}}\|_2 \tag{65}$$

$$\le O\left(\psi\sqrt{\frac{k\log\frac{Lr}{\delta}}{m}}\right)(\|\tilde{\mathbf{x}} - \mu\bar{\mathbf{x}}\|_2 + 2\delta), \tag{66}$$

where (66) uses (61) and the triangle inequality.

2. From Lemma 8, similarly to (63), and applying the union bound, we obtain that for *all* $i \in [l]$ with corresponding $\xi_i > 0$ and *all* $(\mathbf{t}_{i-1}, \mathbf{t}_i)$ pairs in $G(M_{i-1}) \times G(M_i)$, if $m = \Omega(\max_i \xi_i^2)$, then with probability at least $1 - \sum_{i=1}^{l}|M_{i-1}| \cdot |M_i|e^{-\frac{\xi_i^2}{2}}$,

$$\left\langle \frac{1}{m}\mathbf{A}^T(\bar{\mathbf{y}} - \mu\mathbf{A}\bar{\mathbf{x}}), \mathbf{t}_i - \mathbf{t}_{i-1}\right\rangle \le \frac{\xi_i\psi}{\sqrt{m}}\|\mathbf{t}_i - \mathbf{t}_{i-1}\|_2. \tag{67}$$

Since (59) gives $\log(|M_i| \cdot |M_{i-1}|) \le 2ik + 2k\log\frac{4Lr}{\delta}$, if we set $\xi_i = C'\sqrt{ik + k\log\frac{Lr}{\delta}}$ with $C'$ sufficiently large, we obtain

$$\sum_{i=1}^{l}|M_{i-1}| \cdot |M_i|e^{-\frac{\xi_i^2}{2}} = \sum_{i=1}^{l}e^{-\Omega(ik+k\log\frac{Lr}{\delta})} = e^{-\Omega(k\log\frac{Lr}{\delta})}\sum_{i=1}^{l}e^{-\Omega(ik)} = e^{-\Omega(k\log\frac{Lr}{\delta})}. \tag{68}$$

Recall that $\|\tilde{\mathbf{x}}_i - \tilde{\mathbf{x}}_{i-1}\|_2 \leq \frac{\delta}{2^{i-1}}$ for all $i \in [l]$. Then, we obtain that if $m = \Omega\left(k\left(l + \log\frac{Lr}{\delta}\right)\right)$, with probability $1 - e^{-\Omega\left(k\log\frac{Lr}{\delta}\right)}$, the second term in (62) can be upper bounded by

$$\sum_{i=1}^{l}\left\langle \frac{1}{m}\mathbf{A}^T(\bar{\mathbf{y}} - \mu\mathbf{A}\bar{\mathbf{x}}), \tilde{\mathbf{x}}_i - \tilde{\mathbf{x}}_{i-1}\right\rangle \leq \frac{\psi}{\sqrt{m}}\sum_{i=1}^{l}\xi_i\|\tilde{\mathbf{x}}_i - \tilde{\mathbf{x}}_{i-1}\|_2 \tag{69}$$

$$\leq C'\psi\sum_{i=1}^{l}\sqrt{\frac{ik + k\log\frac{Lr}{\delta}}{m}} \times \frac{\delta}{2^{i-1}} \tag{70}$$

$$\leq C'\psi\delta\sqrt{\frac{k}{m}}\sum_{i=1}^{l}\frac{\sqrt{i} + \sqrt{\log\frac{Lr}{\delta}}}{2^{i-1}} \tag{71}$$

$$= O\left(\psi\delta\sqrt{\frac{k\log\frac{Lr}{\delta}}{m}}\right), \tag{72}$$

where (70) substitutes the choice of $\xi_i$, (71) uses $\sqrt{a+b} \leq \sqrt{a} + \sqrt{b}$, and (72) uses the assumption $Lr = \Omega(\delta n)$ and the fact that $\sum_{i=1}^{\infty}\frac{\sqrt{i}}{2^{i-1}}$ is finite.

3. With $m = \Omega\left(k\log\frac{Lr}{\delta}\right)$, if we set $t = \Omega(k\log\frac{Lr}{\delta})$ in Lemma 7, we obtain with probability $1 - e^{-\Omega\left(k\log\frac{Lr}{\delta}\right)}$ that

$$\left\|\frac{1}{m}\mathbf{A}^T(\bar{\mathbf{y}} - \mu\mathbf{A}\bar{\mathbf{x}})\right\|_{\infty} \leq O\left(\psi\sqrt{\frac{k\log\frac{Lr}{\delta}}{m}}\right). \tag{73}$$

Then, setting $l = \lceil\log_2 n\rceil$, with probability $1 - e^{-\Omega\left(k\log\frac{Lr}{\delta}\right)}$, the third term in (62) can be upper bounded as follows:

$$\left\langle\frac{1}{m}\mathbf{A}^T(\bar{\mathbf{y}} - \mu\mathbf{A}\bar{\mathbf{x}}), \tilde{\mathbf{x}} - \tilde{\mathbf{x}}_l\right\rangle \leq \left\|\frac{1}{m}\mathbf{A}^T(\bar{\mathbf{y}} - \mu\mathbf{A}\bar{\mathbf{x}})\right\|_{\infty}\|\tilde{\mathbf{x}} - \tilde{\mathbf{x}}_l\|_1 \tag{74}$$

$$\leq O\left(\psi\sqrt{\frac{k\log\frac{Lr}{\delta}}{m}}\right)\sqrt{n}\|\tilde{\mathbf{x}} - \tilde{\mathbf{x}}_l\|_2 \tag{75}$$

$$\leq O\left(\psi\sqrt{\frac{k\log\frac{Lr}{\delta}}{m}}\right)\sqrt{n} \times \frac{\delta}{2^l} \tag{76}$$

$$= O\left(\psi\delta\sqrt{\frac{k\log\frac{Lr}{\delta}}{m}}\right), \tag{77}$$

where (74) uses Hölder's inequality, (75) uses $\|\mathbf{v}\|_1 \leq \sqrt{n}\|\mathbf{v}\|_2$ for $\mathbf{v} \in \mathbb{R}^n$, (76) uses the definition of $\tilde{\mathbf{x}}_l$, and (77) uses $l = \lceil\log_2 n\rceil$.

By the assumption $Lr = \Omega(\delta n)$, the choice $l = \lceil\log_2 n\rceil$ leads to $m = \Omega\left(k\left(l + \log\frac{Lr}{\delta}\right)\right) = \Omega\left(k\log\frac{Lr}{\delta}\right)$. Substituting (66), (72), and (77) into (62), we obtain that when $m = \Omega\left(k\log\frac{Lr}{\delta}\right)$, with probability $1 - e^{-\Omega\left(k\log\frac{Lr}{\delta}\right)}$,

$$\left\langle\frac{1}{m}\mathbf{A}^T(\bar{\mathbf{y}} - \mu\mathbf{A}\bar{\mathbf{x}}), \tilde{\mathbf{x}} - \mu\bar{\mathbf{x}}\right\rangle \leq O\left(\psi\sqrt{\frac{k\log\frac{Lr}{\delta}}{m}}\right)\|\tilde{\mathbf{x}} - \mu\bar{\mathbf{x}}\|_2 + O\left(\delta\psi\sqrt{\frac{k\log\frac{Lr}{\delta}}{m}}\right). \tag{78}$$

This completes the proof of Lemma 3.

## C  Omitted Proofs from Section 4 (Other Extensions)

### C.1  Proof Outline for Corollary 2 (Bounded Sparse Vectors)

For fixed $\nu > 0$, let $\mathcal{S}_\nu := \Sigma_k^n \cap \nu B_2^n$, where $\Sigma_k^n$ represents the set of $k$-sparse vectors in $\mathbb{R}^n$. We know that for any $\delta > 0$, there exists a $\delta$-net $\mathcal{M}_\nu$ of $\mathcal{S}_\nu$ with $|\mathcal{M}_\nu| \leq \binom{n}{k}\left(\frac{\nu}{\delta}\right)^k \leq \left(\frac{en\nu}{k\delta}\right)^k =$

$\exp\left(O\left(k\log\frac{\nu n}{\delta k}\right)\right)$ [1]. Using this observation and following the proof of Theorem 1, we can derive the Corollary 2 for the case that the signal comes from the set of bounded $k$-sparse vectors.

## C.2 Proof of Corollary 3 (General Covariance Matrices)

We can write $\mathbf{a}_i$ as $\mathbf{a}_i = \sqrt{\boldsymbol{\Sigma}}\mathbf{b}_i$ with $\mathbf{b}_i \sim \mathcal{N}(\mathbf{0}, \mathbf{I}_n)$. Letting[5] $\mathbf{A} = [\mathbf{a}_1^T; \mathbf{a}_2^T; \ldots; \mathbf{a}_m^T] \in \mathbb{R}^{m \times n}$ and $\mathbf{B} = [\mathbf{b}_1^T; \ldots; \mathbf{b}_m^T] \in \mathbb{R}^{m \times n}$, we have

$$\hat{\mathbf{x}} = \arg\min_{\mathbf{x}\in\mathcal{K}} \|\mathbf{y} - \mathbf{A}\mathbf{x}\|_2$$

$$\Leftrightarrow \hat{\mathbf{x}} = \arg\min_{\mathbf{x}\in\mathcal{K}} \|\mathbf{y} - \mathbf{B}\sqrt{\boldsymbol{\Sigma}}\mathbf{x}\|_2 \tag{79}$$

$$\Leftrightarrow \sqrt{\boldsymbol{\Sigma}}\hat{\mathbf{x}} = \arg\min_{\mathbf{x}\in\sqrt{\boldsymbol{\Sigma}}\mathcal{K}} \|\mathbf{y} - \mathbf{B}\mathbf{x}\|_2. \tag{80}$$

Define $\hat{G}$ as $\hat{G}(\mathbf{z}) = \sqrt{\boldsymbol{\Sigma}}G(\mathbf{z})$ for all $\mathbf{z} \in B_2^k(r)$. Then, it is straightforward to establish that $\hat{G}$ is $\hat{L}$-Lipschitz with $\hat{L} = \|\boldsymbol{\Sigma}\|_{2\to2}^{\frac{1}{2}}L$. In addition, we have $\mathbf{y} = f(\mathbf{A}\mathbf{x}^*) = f(\mathbf{B}\sqrt{\boldsymbol{\Sigma}}\mathbf{x}^*)$, $\|\sqrt{\boldsymbol{\Sigma}}\mathbf{x}^*\|_2 = 1$ and $\mu(\sqrt{\boldsymbol{\Sigma}}\mathbf{x}^*) \in \sqrt{\boldsymbol{\Sigma}}\mathcal{K} = \hat{G}(B_2^k(r))$. Applying Theorem 1, we obtain that when $\|\boldsymbol{\Sigma}\|_{2\to2}^{\frac{1}{2}}Lr = \Omega(\epsilon\psi n)$ and $m = \Omega\left(\frac{k}{\epsilon^2}\log\frac{\|\boldsymbol{\Sigma}\|_{2\to2}^{\frac{1}{2}}Lr}{\epsilon\psi}\right)$, with probability $1 - e^{-\Omega(\epsilon^2 m)}$,

$$\|\sqrt{\boldsymbol{\Sigma}}\hat{\mathbf{x}} - \mu\sqrt{\boldsymbol{\Sigma}}\mathbf{x}^*\|_2 \le \psi\epsilon + \tau, \tag{81}$$

as desired.

## D  Alternative Model for Binary Measurements

For binary observations, the following measurement model is considered in various works [4, 24, 43, 44]: The response variables, $y_i \in \{-1, 1\}, i \in [m]$, are drawn independently at random according to some distribution satisfying

$$\mathbb{E}[y_i|\mathbf{a}_i] = \theta(\mathbf{a}_i^T\mathbf{x}^*), \tag{82}$$

for some deterministic function $\theta$ with $-1 \le \theta(z) \le 1$. In this section, we provide a result related to Theorem 1 for this model, again considering the case that $\mathbf{a}_i \sim \mathcal{N}(\mathbf{0}, \mathbf{I}_n)$ and $\mathbf{x}^* \in \mathcal{K} \cap \mathcal{S}^{n-1}$ with $\mathcal{K} = G(B_2^k(r))$ for some $L$-Lipschitz generative model $G$.

The model (82) is a special case of (3) in which $f(g) \in \{-1, 1\}$ and $\mathbb{E}[f(g)] = \theta(g)$. Using this interpretation and the tower property of expectation, we readily find that

$$\mu = \mathbb{E}[\mathbb{E}[f(g)g \,|\, g]] = \mathbb{E}[\theta(g)g] \tag{83}$$

with $g \sim \mathcal{N}(0, 1)$. In addition, we have for any $i \in [m]$ that

$$\mathbb{E}[y_i\mathbf{a}_i^T\mathbf{x}^*] = \mathbb{E}[\mathbb{E}[y_i\mathbf{a}_i^T\mathbf{x}^*|\mathbf{a}_i]] = \mathbb{E}[(\mathbf{a}_i^T\mathbf{x}^*)\theta(\mathbf{a}_i^T\mathbf{x}^*)] = \mu, \tag{84}$$

and it is straightforward to show that [43, Lemma 4]

$$\mathbb{E}[y_i\mathbf{a}_i] = \mu\mathbf{x}^*. \tag{85}$$

Let $\tilde{\mathbf{y}} \in \{-1, 1\}^m$ be a vector of corrupted observations satisfying $\frac{1}{\sqrt{m}}\|\mathbf{y} - \tilde{\mathbf{y}}\|_2 \le \tau$. To derive an estimator for $\mathbf{x}^*$, we seek $\hat{\mathbf{x}}$ *maximizing* $\tilde{\mathbf{y}}^T(\mathbf{A}\mathbf{x})$ over $\mathbf{x} \in \mathcal{K} = G(B_2^k(r))$, i.e.,

$$\hat{\mathbf{x}} := \arg\max_{\mathbf{x}\in\mathcal{K}} \tilde{\mathbf{y}}^T(\mathbf{A}\mathbf{x}). \tag{86}$$

As was done in previous works such as [24, 43], we assume that the considered low-dimensional set is contained in the unit Euclidean ball, i.e., $\mathcal{K} \subseteq B_2^n$. In this section, we establish the following theorem, which is similar to Theorem 1. Although the ideas are similar, the model assumptions and the algorithms used are slightly different, so the results are both of interest.

**Theorem 3.** *Consider any* $\mathbf{x}^* \in \mathcal{K} \cap \mathcal{S}^{n-1}$ *with* $\mathcal{K} = G(B_2^k(r)) \subseteq B_2^n$ *for some* L*-Lipschitz generative model* $G : B_2^k(r) \to \mathbb{R}^n$, *along with* $\mathbf{y}$ *generated from the model* (82) *with* $\mathbf{a}_i \overset{i.i.d.}{\sim} \mathcal{N}(\mathbf{0}, \mathbf{I}_n)$, *and an arbitrary corrupted vector* $\tilde{\mathbf{y}}$ *with* $\frac{1}{\sqrt{m}}\|\tilde{\mathbf{y}} - \mathbf{y}\|_2 \le \tau$. *For any* $\epsilon > 0$, *if* $Lr = \Omega(\epsilon n)$ *and* $m = \Omega\left(\frac{k}{\epsilon^2}\log\frac{Lr}{\epsilon}\right)$, *then with probability* $1 - e^{-\Omega(\epsilon^2 m)}$, *any solution* $\hat{\mathbf{x}}$ *to* (86) *satisfies*

$$\|\mathbf{x}^* - \hat{\mathbf{x}}\|_2 \le \frac{\epsilon + \tau}{\mu}. \tag{87}$$

The proof is mostly similar to that of Theorem 1, so we only outline the differences in the following.

### D.1 Auxiliary Results

In the remainder of this appendix, we assume that the binary vector $\mathbf{y}$ is generated according to (82). Note that for binary measurements, the relevant random variables are sub-Gaussian, and thus we only need concentration inequalities for sub-Gaussian random variables, instead of those for sub-exponential random variables. According to [35, Proposition 5.10], we have the following concentration inequality for sub-Gaussian random variables.

**Lemma 9.** (Hoeffding-type inequality [35, Proposition 5.10]) *Let $X_1, \ldots, X_N$ be independent zero-mean sub-Gaussian random variables, and let $K = \max_i \|X_i\|_{\psi_2}$. Then, for any $\boldsymbol{\alpha} = [\alpha_1, \alpha_2, \ldots, \alpha_N]^T \in \mathbb{R}^N$ and any $t \geq 0$, it holds that*

$$\mathbb{P}\left(\Big|\sum_{i=1}^N \alpha_i X_i\Big| \geq t\right) \leq \exp\left(1 - \frac{ct^2}{K^2 \|\boldsymbol{\alpha}\|_2^2}\right), \tag{88}$$

*where $c > 0$ is a constant.*

By Lemma 9 and the equality $\mathbb{E}[y_i \mathbf{a}_i] = \lambda \mathbf{x}^*$, we arrive at the following lemma, which is similar to Lemma 7.

**Lemma 10.** [43, Lemma 3] *With probability at least $1 - e^{1-t}$, we have*

$$\left\|\frac{1}{m}\mathbf{A}^T \mathbf{y} - \lambda \mathbf{x}^*\right\|_\infty \leq c\sqrt{\frac{t + \log n}{m}} \tag{89}$$

*for a certain constant $c > 0$.*

The following lemma is proved similarly to Lemma 8, so the details are omitted.

**Lemma 11.** *For any $\mathbf{u} \in \mathbb{R}^n$, the random variable $U := \left\langle \frac{1}{m}\mathbf{A}^T\mathbf{y} - \lambda \mathbf{x}^*, \mathbf{u}\right\rangle$ is sub-Gaussian with zero mean. Moreover, for any $\xi > 0$, with probability $1 - e^{-\Omega(\xi^2)}$, we have*

$$|U| \leq \frac{\xi \|\mathbf{u}\|_2}{\sqrt{m}}. \tag{90}$$

Finally, based on Lemmas 10 and 11, and by using a chain of nets similarly to (59)–(60), we derive the following analog of Lemma 3, whose proof is again omitted due to similarity. Note that Lemmas 10 and 11 are only used to derive Lemma 12, and they are not directly used in the proof of Theorem 3.

**Lemma 12.** *For any $\delta > 0$, if $Lr = \Omega(\delta n)$ and $m = \Omega\big(k \log \frac{Lr}{\delta}\big)$, then with probability $1 - e^{-\Omega(k \log \frac{Lr}{\delta})}$, it holds that*

$$\left\langle \frac{1}{m}\mathbf{A}^T\mathbf{y} - \lambda \mathbf{x}^*, \hat{\mathbf{x}} - \mathbf{x}^*\right\rangle \leq O\left(\sqrt{\frac{k \log \frac{Lr}{\delta}}{m}}\right) \|\mathbf{x}^* - \hat{\mathbf{x}}\|_2 + O\left(\delta\sqrt{\frac{k \log \frac{Lr}{\delta}}{m}}\right). \tag{91}$$

### D.2 Proof Outline for Theorem 3

Because $\hat{\mathbf{x}}$ maximizes $\tilde{\mathbf{y}}^T(\mathbf{A}\mathbf{x})$ within $\mathcal{K}$ and we assume $\mathbf{x}^* \in \mathcal{K}$, we obtain

$$\tilde{\mathbf{y}}^T(\mathbf{A}\hat{\mathbf{x}}) \geq \tilde{\mathbf{y}}^T(\mathbf{A}\mathbf{x}^*), \tag{92}$$

which gives the following after some simple manipulations:

$$\langle \mu\mathbf{x}^*, \mathbf{x}^* - \hat{\mathbf{x}}\rangle \leq \left\langle \frac{1}{m}\mathbf{A}^T\tilde{\mathbf{y}} - \mu\mathbf{x}^*, \hat{\mathbf{x}} - \mathbf{x}^*\right\rangle. \tag{93}$$

Using $\|\hat{\mathbf{x}}\|_2 \leq 1$ and $\|\mathbf{x}^*\|_2 = 1$, we derive a lower bound for $\langle \mu\mathbf{x}^*, \mathbf{x}^* - \hat{\mathbf{x}}\rangle$, i.e.,

$$\frac{\mu}{2}\|\hat{\mathbf{x}} - \mathbf{x}^*\|_2^2 \leq \langle \mu\mathbf{x}^*, \mathbf{x}^* - \hat{\mathbf{x}}\rangle. \tag{94}$$

Once this result is in place, the analysis proceeds similarly to that of Theorem 1: Similar to (19), we derive an upper bound for the adversarial noise term, and using Lemma 12 (which is similar to Lemma 3) to derive the following analog of (20):

$$\left\langle \frac{1}{m}\mathbf{A}^T\tilde{\mathbf{y}} - \mu\mathbf{x}^*, \hat{\mathbf{x}} - \mathbf{x}^*\right\rangle \leq \left(\tau + \sqrt{\frac{k \log \frac{Lr}{\delta}}{m}}\right)\|\mathbf{x}^* - \hat{\mathbf{x}}\|_2 + O\left(\tau\delta + \delta\sqrt{\frac{k \log \frac{Lr}{\delta}}{m}}\right). \tag{95}$$

Combining (94) and (95), and using similar steps to those following (23) in the proof of Theorem 1, we derive the desired upper bound for $\|\mathbf{x}^* - \hat{\mathbf{x}}\|_2$. The details are omitted to avoid repetition.

# E Relation to the Gaussian Mean Width

The (global) Gaussian mean width (GMW) of a set $\mathcal{K}$ is defined as

$$\omega(\mathcal{K}) := \mathbb{E}\left[\sup_{\mathbf{x}\in\mathcal{K}-\mathcal{K}} \langle\mathbf{g},\mathbf{x}\rangle\right], \tag{96}$$

where $\mathcal{K}-\mathcal{K} := \{\mathbf{s}-\mathbf{t} : \mathbf{s}\in\mathcal{K}, \mathbf{t}\in\mathcal{K}\}$ and $\mathbf{g}\sim\mathcal{N}(\mathbf{0},\mathbf{I}_n)$. The GMW of $\mathcal{K}$ is a geometric parameter, and is useful for understanding the effective dimension of $\mathcal{K}$ in estimation problems. In various related works such as [25, 26], the sample complexity derived depends directly on the GMW or its local variants. For example, if $\mathcal{K}\subseteq\mathbb{R}^n$ is compact and star shaped, then by [26, Eq. (2.1)], $m = O\left(\frac{\omega(\mathcal{K})^2}{\epsilon^4}\right)$ measurements suffice for $\epsilon$-accurate recovery.

According to [24], the GMW satisfies the following properties:

1. If $\mathcal{K} = B_2^n$ or $\mathcal{K} = \mathcal{S}^{n-1}$, then $\omega(\mathcal{K}) = \mathbb{E}[\|\mathbf{g}\|_2] \leq \left(\mathbb{E}\left[\|\mathbf{g}\|_2^2\right]\right)^{1/2} = \sqrt{n}$;
2. If $\mathcal{K}$ is a finite set contained in $B_2^n$, then $\omega(\mathcal{K}) \leq C\sqrt{\log|\mathcal{K}|}$.

Using these observations, we obtain the following lemma.

**Lemma 13.** *Fix $r > 0$, and let $G$ be an $L$-Lipschitz generative model with $Lr = \Omega(1)$, and let $\mathcal{K} = G(B_2^k(r)) \subseteq B_2^n$. Then, we have*

$$\omega(\mathcal{K})^2 = \Theta\left(k\log\frac{Lr\sqrt{n}}{\sqrt{k}}\right). \tag{97}$$

*Proof.* As we stated in (59), for any $\delta > 0$, there exists a set $M \subseteq B_2^k(r)$ being a $\frac{\delta}{L}$-net of $B_2^k(r)$ with $\log|M| \leq k\log\frac{4Lr}{\delta}$, and $G(M)$ is a $\delta$-net of $\mathcal{K}$. For any $\mathbf{x}\in\mathcal{K}-\mathcal{K}$, there exists $\mathbf{s}\in G(M)-G(M)$ with $\|\mathbf{x}-\mathbf{s}\|_2 \leq 2\delta$; hence,

$$\langle\mathbf{g},\mathbf{x}\rangle \leq \langle\mathbf{g},\mathbf{s}\rangle + \|\mathbf{g}\|_2\|\mathbf{x}-\mathbf{s}\|_2 \leq \langle\mathbf{g},\mathbf{s}\rangle + 2\delta\|\mathbf{g}\|_2. \tag{98}$$

As a result, we have

$$\omega(\mathcal{K}) = \mathbb{E}\left[\sup_{\mathbf{x}\in\mathcal{K}-\mathcal{K}}\langle\mathbf{g},\mathbf{x}\rangle\right] \tag{99}$$

$$\leq \omega(G(M)) + 2\delta\mathbb{E}[\|\mathbf{g}\|_2] \tag{100}$$

$$\leq C\sqrt{k\log\frac{4Lr}{\delta}} + 2\delta\sqrt{n}. \tag{101}$$

By a similar argument, we also have

$$\omega(\mathcal{K}) \geq C\sqrt{k\log\frac{4Lr}{\delta}} - 2\delta\sqrt{n}. \tag{102}$$

Setting $\delta = \sqrt{\frac{k}{n}}$ and applying the assumption $Lr = \Omega(1)$, we obtain the desired result. $\qquad\square$

We emphasize that the above analysis assumes that $G(B_2^k(r)) \subseteq B_2^n$, and in the absence of such an assumption, the Gaussian mean width $\omega(\mathcal{K})$ will generally grow linearly with the radius.

Returning to the sample complexity $m = O\left(\frac{k}{\epsilon^2}\log\frac{Lr}{\psi\epsilon}\right)$ in Theorem 1, we find that this reduces to $m = O\left(\frac{\omega(\mathcal{K})^2}{\epsilon^2}\right)$ in broad scaling regimes. For instance, this is the case when $\psi$ is constant, $Lr = n^{\Omega(1)}$ (as is typical for neural networks [2]), and $\epsilon$ decays no faster than polynomially in $n$.

# F Local Embedding Property (LEP) for the 1-bit Model

For $\mathbf{v}, \mathbf{v}' \in \mathbb{R}^m$, let $d_H(\mathbf{v}, \mathbf{v}') := \frac{1}{m}\sum_{i=1}^m \mathbf{1}\{v_i \neq v_i'\}$ denote the (normalized) Hamming distance. Note that when $f(x) = \text{sign}(x)$, we obtain $\mu = \mathbb{E}[f(g)g] = \sqrt{\frac{2}{\pi}}$ and $\psi = 1$. We have the following lemma, which essentially states that for all $\mathbf{x}, \mathbf{s} \in \mathcal{S}^{n-1}$, if $\mathbf{x}$ is close to $\mathbf{s}$ in $\ell_2$-norm, then $\text{sign}(\mathbf{Ax})$ is close to $\text{sign}(\mathbf{As})$ in Hamming distance.

**Lemma 14.** (Adapted from [17, Corollary 2]) *For fixed $\epsilon \in (0,1)$, if $m = \Omega\left(\frac{k}{\epsilon}\log\frac{Lr}{\mu\epsilon^2}\right)$, with probability $1 - e^{-\Omega(\epsilon m)}$, for all $\mathbf{x}_1, \mathbf{x}_2 \in \mathcal{S}^{n-1}$ with $\mu_1\mathbf{x}_1, \mu_2\mathbf{x}_2 \in \mathcal{K}$, where $\mu_1, \mu_2 = \Theta(\mu)$, it holds that*

$$\|\mathbf{x}_1 - \mathbf{x}_2\|_2 \le \epsilon \Rightarrow \mathrm{d}_{\mathrm{H}}(\mathrm{sign}(\mathbf{Ax}_1), \mathrm{sign}(\mathbf{Ax}_2)) \le O(\epsilon). \tag{103}$$

Note that each entry of $|\mathrm{sign}(\mathbf{Ax}_1) - \mathrm{sign}(\mathbf{Ax}_2)|$ is either 2 or 0. Hence, if (103) is satisfied, we have

$$\frac{1}{\sqrt{m}}\|\mathrm{sign}(\mathbf{Ax}_1) - \mathrm{sign}(\mathbf{Ax}_2)\|_2 = 2\sqrt{\mathrm{d}_{\mathrm{H}}(\mathrm{sign}(\mathbf{Ax}_1), \mathrm{sign}(\mathbf{Ax}_2))} \le O(\sqrt{\epsilon}). \tag{104}$$

That is, setting $\beta = \frac{1}{2}$, we have that $f(x) = \mathrm{sign}(x)$ satisfies Assumption 1 in Section 5 with $M_{\mathrm{LEP}}(\delta, \beta) = O\left(\frac{k}{\delta}\log\frac{Lr}{\mu\delta^2}\right)$ and $P_{\mathrm{LEP}}(\delta, \beta) = 1 - e^{-\Omega(\delta m)}$.

# G  Proof of Theorem 2 (Uniform Recovery)

We briefly repeat the argument at the start of the proof of Lemma 3: For fixed $\delta \in (0,1)$ and a positive integer $l$, let $M = M_0 \subseteq M_1 \subseteq \ldots \subseteq M_l$ be a chain of nets of $B_2^k(r)$ such that $M_i$ is a $\frac{\delta_i}{L}$-net with $\delta_i = \frac{\delta}{2^i}$. There exists such a chain of nets with

$$\log|M_i| \le k\log\frac{4Lr}{\delta_i}. \tag{105}$$

By the $L$-Lipschitz assumption on $G$, we have for any $i \in [l]$ that $G(M_i)$ is a $\delta_i$-net of $G(B_2^k(r))$. We write $\mu\mathbf{x}^*$ and $\hat{\mathbf{x}}$ as

$$\mu\mathbf{x}^* = (\mu\mathbf{x}^* - \mu\mathbf{x}_l^*) + (\mu\mathbf{x}_l^* - \mu\mathbf{x}_{l-1}^*) + \ldots + (\mu\mathbf{x}_1^* - \mu\mathbf{x}_0^*) + \mu\mathbf{x}_0^*, \tag{106}$$

$$\hat{\mathbf{x}} = (\hat{\mathbf{x}} - \hat{\mathbf{x}}_l) + (\hat{\mathbf{x}}_l - \hat{\mathbf{x}}_{l-1}) + \ldots + (\hat{\mathbf{x}}_1 - \hat{\mathbf{x}}_0) + \hat{\mathbf{x}}_0, \tag{107}$$

where $\hat{\mathbf{x}}_i, \mu\mathbf{x}_i^* \in G(M_i)$ for all $i \in [l]$, and $\|\hat{\mathbf{x}} - \hat{\mathbf{x}}_l\|_2 \le \frac{\delta}{2^l}$, $\|\mu\mathbf{x}^* - \mu\mathbf{x}_l^*\|_2 \le \frac{\delta}{2^l}$, and $\|\hat{\mathbf{x}}_i - \hat{\mathbf{x}}_{i-1}\|_2 \le \frac{\delta}{2^{i-1}}$, $\|\mu\mathbf{x}_i^* - \mu\mathbf{x}_{i-1}^*\|_2 \le \frac{\delta}{2^{i-1}}$ for all $i \in [l]$. Therefore, the triangle inequality gives

$$\|\hat{\mathbf{x}} - \hat{\mathbf{x}}_0\|_2 < 2\delta, \quad \|\mu\mathbf{x}^* - \mu\mathbf{x}_0^*\|_2 < 2\delta. \tag{108}$$

In analogy with (62), we write

$$\left\langle \frac{1}{m}\mathbf{A}^T(\tilde{\mathbf{y}} - \mu\mathbf{Ax}^*), \hat{\mathbf{x}} - \mu\mathbf{x}^* \right\rangle$$

$$= \left\langle \frac{1}{m}\mathbf{A}^T(\tilde{\mathbf{y}} - \mathbf{y}), \hat{\mathbf{x}} - \mu\mathbf{x}^* \right\rangle + \left\langle \frac{1}{m}\mathbf{A}^T\left(\mathbf{y} - f\left(\mathbf{A}\frac{\mathbf{x}_0^*}{\|\mathbf{x}_0^*\|_2}\right)\right), \hat{\mathbf{x}} - \mu\mathbf{x}^* \right\rangle$$

$$+ \left\langle \frac{1}{m}\mathbf{A}^T\left(f\left(\mathbf{A}\frac{\mathbf{x}_0^*}{\|\mathbf{x}_0^*\|_2}\right) - \mu\mathbf{A}\frac{\mathbf{x}_0^*}{\|\mathbf{x}_0^*\|_2}\right), \hat{\mathbf{x}} - \mu\mathbf{x}^* \right\rangle + \left\langle \frac{1}{m}\mathbf{A}^T\mu\mathbf{A}\left(\frac{\mathbf{x}_0^*}{\|\mathbf{x}_0^*\|_2} - \mathbf{x}^*\right), \hat{\mathbf{x}} - \mu\mathbf{x}^* \right\rangle \tag{109}$$

and proceed by deriving uniform upper bounds for the four terms in (109) separately. In the following, we assume that $m = \Omega\left(k\log\frac{Lr}{\delta}\right)$; we will later choose $\delta$ such that this reduces to $m = \Omega\left(k\log\frac{Lr}{\epsilon}\right)$, as in the theorem statement.

1. A uniform upper bound for $\left\langle \frac{1}{m}\mathbf{A}^T(\tilde{\mathbf{y}} - \mathbf{y}), \hat{\mathbf{x}} - \mu\mathbf{x}^* \right\rangle$: Recall that from (19), we have

$$\left\langle \frac{1}{m}\mathbf{A}^T(\mathbf{y} - \tilde{\mathbf{y}}), \hat{\mathbf{x}} - \mu\mathbf{x}^* \right\rangle \le \left\|\frac{1}{\sqrt{m}}(\mathbf{y} - \tilde{\mathbf{y}})\right\|_2 \times \left\|\frac{1}{\sqrt{m}}\mathbf{A}(\hat{\mathbf{x}} - \mu\mathbf{x}^*)\right\|_2 \tag{110}$$

$$\le \tau O(\|\hat{\mathbf{x}} - \mu\mathbf{x}^*\|_2 + \delta). \tag{111}$$

This inequality holds uniformly for all $\hat{\mathbf{x}}, \mu\mathbf{x}^* \in \mathcal{K}$, since it is based on the uniform result in Lemma 2.

2. A uniform upper bound for $\left\langle \frac{1}{m}\mathbf{A}^T\left(\mathbf{y} - f\left(\mathbf{A}\frac{\mathbf{x}_0^*}{\|\mathbf{x}_0^*\|_2}\right)\right), \hat{\mathbf{x}} - \mu\mathbf{x}^* \right\rangle$: From (108), we have $\|\mathbf{x}^* - \mathbf{x}_0^*\|_2 \le \frac{2\delta}{\mu}$. Because $\|\mathbf{x}^*\|_2 = 1$ and $\|\mathbf{x}^* - \mathbf{x}_0^*\|_2 \ge |\|\mathbf{x}_0^*\|_2 - \|\mathbf{x}^*\|_2|$, we obtain

$$\left\|\mathbf{x}_0^* - \frac{\mathbf{x}_0^*}{\|\mathbf{x}_0^*\|_2}\right\|_2 = \left\|\frac{\mathbf{x}_0^*(\|\mathbf{x}_0^*\|_2 - 1)}{\|\mathbf{x}_0^*\|_2}\right\|_2 \le |\|\mathbf{x}_0^*\|_2 - 1| \le \frac{2\delta}{\mu}, \tag{112}$$

and the triangle inequality gives

$$\left\| \mathbf{x}^* - \frac{\mathbf{x}_0^*}{\|\mathbf{x}_0^*\|_2} \right\|_2 \le \frac{4\delta}{\mu}. \tag{113}$$

If we choose $\delta \le c'\mu$ for sufficiently small $c'$, then we obtain $\|\mathbf{x}_0^*\|_2 \in [1 - \eta_0, 1 + \eta_0]$ for arbitrarily small $\eta_0$, which implies that $c\frac{\mathbf{x}_0^*}{\|\mathbf{x}_0^*\|_2} \in \mathcal{K}$ for some $c \in [\mu - \eta, \mu + \eta]$ and arbitrarily small $\eta > 0$ (since $\mu\mathbf{x}_0^* \in \mathcal{K}$ and $\mu = \Theta(1)$). Hence, considering Assumption 1, we observe that the high-probability LEP condition (34) therein (along with $\mu = \Theta(1)$) implies

$$\frac{1}{\sqrt{m}} \left\| \mathbf{y} - f\left( \mathbf{A}\frac{\mathbf{x}_0^*}{\|\mathbf{x}_0^*\|_2} \right) \right\|_2 = \frac{1}{\sqrt{m}} \left\| f(\mathbf{A}\mathbf{x}^*) - f\left( \mathbf{A}\frac{\mathbf{x}_0^*}{\|\mathbf{x}_0^*\|_2} \right) \right\|_2 \le O\left( \delta^\beta \right), \tag{114}$$

Then, similarly to the derivation of (111), we have that if $m \ge M_{\mathrm{LEP}}(\delta, \beta) + \Omega\left( k \log \frac{Lr}{\delta} \right)$, then with probability $1 - P_{\mathrm{LEP}}(\delta, \beta) - e^{-\Omega(m)}$,

$$\left\langle \frac{1}{m}\mathbf{A}^T \left( \mathbf{y} - f\left( \mathbf{A}\frac{\mathbf{x}_0^*}{\|\mathbf{x}_0^*\|_2} \right) \right), \hat{\mathbf{x}} - \mu\mathbf{x}^* \right\rangle$$

$$\le \left\| \frac{1}{\sqrt{m}} \left( \mathbf{y} - f\left( \mathbf{A}\frac{\mathbf{x}_0^*}{\|\mathbf{x}_0^*\|_2} \right) \right) \right\|_2 \times \left\| \frac{1}{\sqrt{m}}\mathbf{A}(\hat{\mathbf{x}} - \mu\mathbf{x}^*) \right\|_2 \tag{115}$$

$$\le O(\delta^\beta) \times O(\|\hat{\mathbf{x}} - \mu\mathbf{x}^*\|_2 + \delta) \tag{116}$$

$$= O(\delta^\beta \|\hat{\mathbf{x}} - \mu\mathbf{x}^*\|_2 + \delta^{\beta+1}). \tag{117}$$

3. A uniform upper bound for $\left\langle \frac{1}{m}\mathbf{A}^T \left( f\left( \mathbf{A}\frac{\mathbf{x}_0^*}{\|\mathbf{x}_0^*\|_2} \right) - \mu\mathbf{A}\frac{\mathbf{x}_0^*}{\|\mathbf{x}_0^*\|_2} \right), \hat{\mathbf{x}} - \mu\mathbf{x}^* \right\rangle$: For brevity, let
$\mathbf{s}_0 = \frac{1}{m}\mathbf{A}^T \left( f\left( \mathbf{A}\frac{\mathbf{x}_0^*}{\|\mathbf{x}_0^*\|_2} \right) - \mu\mathbf{A}\frac{\mathbf{x}_0^*}{\|\mathbf{x}_0^*\|_2} \right)$. We have

$$\langle \mathbf{s}_0, \hat{\mathbf{x}} - \mu\mathbf{x}^* \rangle = \left\langle \mathbf{s}_0, \hat{\mathbf{x}} - \mu\frac{\mathbf{x}_0^*}{\|\mathbf{x}_0^*\|_2} \right\rangle + \left\langle \mathbf{s}_0, \mu\left( \frac{\mathbf{x}_0^*}{\|\mathbf{x}_0^*\|_2} - \mathbf{x}^* \right) \right\rangle. \tag{118}$$

By Lemma 3 and the union bound over $G(M)$ (for $\mathbf{x}_0^*$), we obtain with probability $1 - |M|e^{-\Omega(k \log \frac{Lr}{\delta})} = 1 - e^{-\Omega(k \log \frac{Lr}{\delta})}$ that

$$\left\langle \mathbf{s}_0, \hat{\mathbf{x}} - \mu\frac{\mathbf{x}_0^*}{\|\mathbf{x}_0^*\|_2} \right\rangle \le O\left( \sqrt{\frac{k \log \frac{Lr}{\delta}}{m}} \right) \left\| \hat{\mathbf{x}} - \mu\frac{\mathbf{x}_0^*}{\|\mathbf{x}_0^*\|_2} \right\|_2 + O\left( \delta\sqrt{\frac{k \log \frac{Lr}{\delta}}{m}} \right) \tag{119}$$

$$\le O\left( \sqrt{\frac{k \log \frac{Lr}{\delta}}{m}} \right) (\|\hat{\mathbf{x}} - \mu\mathbf{x}^*\|_2 + 4\delta) + O\left( \delta\sqrt{\frac{k \log \frac{Lr}{\delta}}{m}} \right) \tag{120}$$

$$= O\left( \sqrt{\frac{k \log \frac{Lr}{\delta}}{m}} \right) \|\hat{\mathbf{x}} - \mu\mathbf{x}^*\|_2 + O\left( \delta\sqrt{\frac{k \log \frac{Lr}{\delta}}{m}} \right), \tag{121}$$

where (120) follows from the triangle inequality and (113). In addition, we have

$$\left\langle \mathbf{s}_0, \mu\left( \frac{\mathbf{x}_0^*}{\|\mathbf{x}_0^*\|_2} - \mathbf{x}^* \right) \right\rangle$$

$$= \left\langle \mathbf{s}_0, \mu\left( \frac{\mathbf{x}_0^*}{\|\mathbf{x}_0^*\|_2} - \mathbf{x}_0^* \right) \right\rangle + \langle \mathbf{s}_0, \mu(\mathbf{x}_l^* - \mathbf{x}^*) \rangle + \sum_{i=1}^{l} \langle \mathbf{s}_0, \mu(\mathbf{x}_{i-1}^* - \mathbf{x}_i^*) \rangle. \tag{122}$$

Then, by Lemma 8 and the union bound over $G(M)$ (for $\mathbf{x}_0^*$), we obtain with probability $1 - e^{-\Omega(k \log \frac{Lr}{\delta})}$ that

$$\left\langle \mathbf{s}_0, \mu\left( \frac{\mathbf{x}_0^*}{\|\mathbf{x}_0^*\|_2} - \mathbf{x}_0^* \right) \right\rangle \le O\left( \sqrt{\frac{k \log \frac{Lr}{\delta}}{m}} \right) \mu \left\| \frac{\mathbf{x}_0^*}{\|\mathbf{x}_0^*\|_2} - \mathbf{x}_0^* \right\|_2 \le O\left( \delta\sqrt{\frac{k \log \frac{Lr}{\delta}}{m}} \right), \tag{123}$$

where the last inequality uses (112). Similar to that in the proof of Lemma 3, we set $l = \lceil \log_2 n \rceil$. By (77), the union bound over $G(M)$ (for $\mathbf{x}_0^*$), and the assumption $\psi = \Theta(1)$, we obtain with probability $1 - e^{-\Omega(k \log \frac{Lr}{\delta})}$ that

$$\langle \mathbf{s}_0, \mu(\mathbf{x}_l^* - \mathbf{x}^*) \rangle \leq O\left( \delta \sqrt{\frac{k \log \frac{Lr}{\delta}}{m}} \right). \tag{124}$$

In addition, by (72) and a union bound over both $G(M)$ and over $G(M_{i-1}) \times G(M_i)$ for all $i \in [l]$, we obtain with probability $1 - e^{-\Omega(k \log \frac{Lr}{\delta})}$ that

$$\sum_{i=1}^{l} \left\langle \mathbf{s}_0, \mu(\mathbf{x}_{i-1}^* - \mathbf{x}_i^*) \right\rangle \leq O\left( \delta \sqrt{\frac{k \log \frac{Lr}{\delta}}{m}} \right). \tag{125}$$

Substituting (121)–(125) into (118), we obtain

$$\langle \mathbf{s}_0, \hat{\mathbf{x}} - \mu \mathbf{x}^* \rangle \leq O\left( \delta + \sqrt{\frac{k \log \frac{Lr}{\delta}}{m}} \right) \|\hat{\mathbf{x}} - \mu \mathbf{x}^*\|_2 + O\left( \delta \sqrt{\frac{k \log \frac{Lr}{\delta}}{m}} \right). \tag{126}$$

4. A uniform upper bound for $\left\langle \frac{1}{m} \mathbf{A}^T \mu \mathbf{A} \left( \frac{\mathbf{x}_0^*}{\|\mathbf{x}_0^*\|_2} - \mathbf{x}^* \right), \hat{\mathbf{x}} - \mu \mathbf{x}^* \right\rangle$: From Lemma 2, we have that when $m = \Omega\left( k \log \frac{Lr}{\delta} \right)$, with probability $1 - e^{-\Omega(m)}$,

$$\left\langle \frac{1}{m} \mathbf{A}^T \mu \mathbf{A} \left( \frac{\mathbf{x}_0^*}{\|\mathbf{x}_0^*\|_2} - \mathbf{x}^* \right), \hat{\mathbf{x}} - \mu \mathbf{x}^* \right\rangle$$

$$\leq \left\| \frac{1}{\sqrt{m}} \mu \mathbf{A} \left( \frac{\mathbf{x}_0^*}{\|\mathbf{x}_0^*\|_2} - \mathbf{x}^* \right) \right\|_2 \left\| \frac{1}{\sqrt{m}} \mathbf{A}(\hat{\mathbf{x}} - \mu \mathbf{x}^*) \right\|_2 \tag{127}$$

$$\leq O(\delta) O(\|\hat{\mathbf{x}} - \mu \mathbf{x}^*\|_2 + \delta) = O\left( \delta \|\hat{\mathbf{x}} - \mu \mathbf{x}^*\|_2 + \delta^2 \right). \tag{128}$$

Having bounded the four terms, we now substitute (111), (117), (126), and (128) into (109), and deduce that if $m \geq M_{\text{LEP}}(\delta, \beta) + \Omega\left( k \log \frac{Lr}{\delta} \right)$, then with probability at least $1 - e^{-\Omega(m)} - P_{\text{LEP}}(\delta, \beta)$, it holds uniformly (in both $\mu \mathbf{x}^*$ and $\hat{\mathbf{x}}$) that

$$\left\langle \frac{1}{m} \mathbf{A}^T (\mathbf{y} - \mu \mathbf{A} \mathbf{x}^*), \hat{\mathbf{x}} - \mu \mathbf{x}^* \right\rangle$$

$$\leq O\left( \tau + \delta^\beta + \sqrt{\frac{k \log \frac{Lr}{\delta}}{m}} \right) \|\hat{\mathbf{x}} - \mu \mathbf{x}^*\|_2 + O\left( \delta \tau + \delta \sqrt{\frac{k \log \frac{Lr}{\delta}}{m}} + \delta^{1+\beta} \right). \tag{129}$$

Then, similarly to (23), we derive that if $m \geq M_{\text{LEP}}(\delta, \beta) + \Omega\left( k \log \frac{Lr}{\delta} \right)$, then with probability at least $1 - e^{-\Omega(m)} - P_{\text{LEP}}(\delta, \beta)$, it holds uniformly that

$$\|\mu \mathbf{x}^* - \hat{\mathbf{x}}\|_2^2 \leq O\left( \tau + \delta^\beta + \sqrt{\frac{k \log \frac{Lr}{\delta}}{m}} \right) \|\hat{\mathbf{x}} - \mu \mathbf{x}^*\|_2 + O\left( \delta \tau + \delta \sqrt{\frac{k \log \frac{Lr}{\delta}}{m}} + \delta^{1+\beta} \right), \tag{130}$$

where we used the fact that $\delta^\beta + \delta = O(\delta^\beta)$, since $\beta \leq 1$.

Considering the parameter $\epsilon$ in the theorem statement, we now set $\delta = \epsilon^{1/\beta}$ (i.e., $\epsilon = \delta^\beta$), meaning that the previous requirement $m = \Omega\left( \frac{k}{\epsilon^2} \log \frac{Lr}{\delta} \right)$ reduces to $m = \Omega\left( \frac{k}{\epsilon^2} \log \frac{Lr}{\epsilon^{1/\beta}} \right) = \Omega\left( \frac{k}{\epsilon^2} \log \frac{Lr}{\epsilon} \right)$. In addition, $\sqrt{\frac{k \log \frac{Lr}{\delta}}{m}} = O(\epsilon)$. Since $\epsilon \leq 1$ and $\beta \leq 1$, we have

$$O\left( \tau + \delta^\beta + \sqrt{\frac{k \log \frac{Lr}{\delta}}{m}} \right) \|\hat{\mathbf{x}} - \mu \mathbf{x}^*\|_2 + O\left( \delta \tau + \delta \sqrt{\frac{k \log \frac{Lr}{\delta}}{m}} + \delta^{1+\beta} \right)$$

$$= O(\tau + \epsilon) \|\hat{\mathbf{x}} - \mu \mathbf{x}^*\|_2 + O\left( \epsilon^{1/\beta} \tau + \epsilon^{1+1/\beta} \right) \tag{131}$$

$$= O(\tau + \epsilon) \|\hat{\mathbf{x}} - \mu \mathbf{x}^*\|_2 + O\left( (\epsilon + \tau)^2 \right). \tag{132}$$

Substituting into (130) and considering two cases depending on which term in (132) is larger, we obtain that if $m \geq M_{\mathrm{LEP}}(\epsilon^{1/\beta}, \beta) + \Omega\big(\frac{k}{\epsilon^2} \log \frac{Lr}{\epsilon}\big)$, then with probability at least $1 - e^{-\Omega(m)} - P_{\mathrm{LEP}}(\epsilon^{1/\beta}, \beta)$, it holds uniformly that

$$\|\mu \mathbf{x}^* - \hat{\mathbf{x}}\|_2 \leq O(\tau + \epsilon). \tag{133}$$

## Footnotes

[5] For matrices $\mathbf{V}_1 \in \mathbb{R}^{F_1 \times N}$ and $\mathbf{V}_2 \in \mathbb{R}^{F_2 \times N}$, we let $[\mathbf{V}_1; \mathbf{V}_2]$ denote the vertical concatenation.