[Reviews · NeurIPS 2020]

Review 1

Summary and Contributions: Summary: This paper considers the generalized Lasso for non-linear measurements of generative priors. Contributions: 1. The authors show non-uniform and uniform recovery under different assumptions on the measurement function. 2. The analysis in this paper recovers known results in non-linear compressed sensing of sparse vectors and generative priors, and also has connections to Gaussian width arguments. ---Edit after rebuttal-- After reading the author feedback and discussing with the other reviewers, I agree that there is sufficient novelty to increase my score. As R4 also points out, the presentation of technical details can be improved, and section 4 should be reorganized. One major limitation that remains is the lack of experiments in this paper, and I do not agree with the authors' claim that experiments are not important in this case. Given that this paper borrows the theoretical framework of [8,23,24,27], the lack of experiments in earlier work does not justify the lack of experiments in this paper. Specifically, the important question about whether using generative models is actually helpful for non-linear observations is still left unanswered. (the authors do get a better theoretical error bound of m^(-0.5) as opposed to m^(-0.25) as in [23,24], but it's not clear how much the additional assumptions influence the theoretical bound). --------------------------------

Strengths: + This work generalizes 1-bit compressed sensing to more general non-linear measurement functions. + To the best of my knowledge, although the analysis borrows from Bora et al and Liu et al, it is sufficiently novel.

Weaknesses: - While the analysis is different from existing work on non-linear measurement functions, the insight gained compared to previous work is quite limited. The assumption that the set of interest is a subset of the unit sphere is a strong assumption, but is standard (and necessary) in traditional problems like 1-bit CS. The additional assumption that the measurement function applied to a sub-Gaussian is also a sub-Gaussian severely limits the set of measurement functions, and in my opinion is not a strong enough generalization of 1-bit CS. For example, the results in this paper cannot be used for phase retrieval because f(x) = |x|^2 will not satisfy the sub-Gaussian assumption. - The connection to Gaussian width is interesting, but is in my opinion again a weak statement. For example, if one were to use Gaussian width arguments to derive results similar to Bora et al, then the number of measurements would grow quadratically with the radius r. This paper avoids this dependence by assuming that the feasible set lies within the unit ball/sphere. - The claim that K-Lasso is more practical than the algorithm in Liu et al is a conjecture at best. The K-Lasso proposed here requires the feasible set to lie on the unit sphere, and it is not clear that a heuristic to achieve this is any better/worse than the heuristic in Liu et al. - The statement of Corollary 1 is very confusing and requires some clarification on the quality of the claimed bound. The constant bar{mu} depends on z*, the corollary requires an additional assumption that mu G(z*)/ ||G(z*)|| lies in the unit sphere, and the sub-Gaussian constant psi also depends on the ground truth G(z*). It's not at all clear when any of these assumptions are satisfied, and how these constants depend on one another. - No experimental evaluation.

Correctness: - Lemma 1 as stated is incorrect. The Gaussian width claimed in (21) would correspond to the Gaussian width of K \cap B_2^n, and the error in the proof can be seen from eqn (100) to (101), where it is implicitly assumed that G(M) is contained in the unit ball. If one were to not assume that K lies in the unit ball, then the Gaussian width would grow linearly with the radius r.

Clarity: While all the statements are clearly written, the authors have included a variety of results that can be confusing without adding much novel content. For example, Section 3.1 to Section 4.5 are all minor variations of Theorem 1, and is confusing to read due to frequent context switching. Additionally, most of these are known results that can be rederived using Theorem 1, and hence do not add much to the paper.

Relation to Prior Work: Yes

Reproducibility: Yes

Additional Feedback:


Review 2

Summary and Contributions: The paper theoretically proves that the solution to the generalized lasso problem with nonlinear observations (also called semi-parametric single index model) and generative prior can recover the truth under certain assumptions.

Strengths: 1. The main contribution of the paper is theorem 1. It proves that the semi-parametric single index model with generative prior can be solved by a constrained least squared optimization by providing an error bound. 2. The paper also considers several variations of the results and compares the Lipschitz function settings with GMW, which is the settings used in previous work. 3. The paper provides a sufficient condition to the uniform recovery error bound.

Weaknesses: 1. The results provided in this paper seem kind of incremental to me. The settings and final error bounds are similar to the ones in [23] and [24], as shown in table 1 in appendix A. The difference is just the assumption for signal set K. In [23][24], K satisfies GMW and in this paper, K is the range of a generative model. However, both settings imply that effective dimension is small, which is a key to the proof. Note the difference between the two settings is actually addressed by lemma 2 (this result is stronger than that the effective dimension is small since the input dimension of the G is small in nature), which is from [2] and proved by using a delta-net cover. It seems to me that the main result of this paper is a combination of the lemma in [2] and the results from [23][24]. 2. The paper also gives the uniform recovery guarantee in section 5, but the result relies on assumption 1, which seems too strong and hard to test. I would expect the authors can give a sufficient condition that is neat and informative. 3. It would be better if the paper also provides some empirical studies.

Correctness: I think the results in the paper make sense and are reasonable but I didn't get into the details of proofs in supplimentary.

Clarity: The paper is generally well written. The statement of the problem and the theorems are clear. I'm actually neutral in judging the paper given its strength and weakness listed above. I might change my scores after seeing the authors' feedback and comments from other reviewers. ---- Update ---- I've read the authors' rebuttal and the comments from other reviewers, and I would like to keep my original rating on this paper.

Relation to Prior Work: The paper discusses other related work in section 1.1, where authors compare the difference in the settings and results with previous related work. Table 1 in appendix A also clearly compares the difference among different papers.

Reproducibility: Yes

Additional Feedback:


Review 3

Summary and Contributions: The paper studies theoretical guarantees for recovering signals from nonlinear random measurements via the so-called K-LASSO. The paper assumes a generative signal model consisting of a Lipschitz function, e.g. a neural network, and bounded inputs. It gives precise bounds on the required number of measurements and shows robustness under adversarial noise. Moreover, the paper for the first time provides a so-called uniform recovery result in this context. These results extend previous contributions in several non-trivial ways.

Strengths: The paper significantly extends the analysis of the K-LASSO from linear to nonlinear measurements. It makes only weak assumptions on the nonlinearity; for instance, it does not assume differentiability and, hence, includes quantized and 1-bit measurements. The K-LASSO is solved via gradient descent and hence consists in an efficient algorithm (accepting that convergence is only verified empirically so far). The proofs are non-trivial.

Weaknesses: I only see minor weaknesses: The paper does not contain any numerical evaluations. These would be useful to understand the performance in practice. In fact, there is no theoretical guarantee (yet) that gradient descent converges to the solution of the K-LASSO, so that numerical evidence in the context of nonlinear measurements would help. I would expect that the authors expand a bit on this point beyond Remark 1.

Correctness: Yes, the claims are correct as far as I can see.

Clarity: The paper is very clear written and well-organized.

Relation to Prior Work: The paper contains a very detailed description of the prior work and describes very clearly the advances with respect to prior contributions.

Reproducibility: Yes

Additional Feedback: A very minor comment about the definition of the Gaussian width in (20). If one defines it with respect to K-K, then the Gaussian width of a unit ball or unit sphere equals 2E ||g||_2 for a standard Gaussian vector (factor 2 missing). The factor 2 can be omitted if one defines it only using K (instead of K-K). It would be good if the author could comment on the condition Lr = \Omega(\epsilon \psi n) in Theorem 1. I understand that it is just a technical condition, which is satisfied in most situations of interest, but the authors may want to say a bit more about this point. Remark 2, line 202: The word "that" should be removed. --------------------------------- Comment after reading author feedback and the other reviews: I still think that this is an excellent contribution and I see no reason to change my evaluation.


Review 4

Summary and Contributions: This paper generalizes result on nonlinear distortion of compressive measurements to generative signal models, such as deep image priors. Results of this flavor already exist for sparse models.

Strengths: The paper provide theoretical results to demonstrate that thee behavior we expect in sparse signal models is also true in generative models. This is comforting, and allows broader use of such models.

Weaknesses: While the results are solid theoretically, they are not surprising. It seems like the paper provides a straightforward extension of existing results in thee references. In addition, given the theoretical nature of the paper, I find it disappointing that most of the proofs (especially for Thm. 1) are in the appendix in the supplemental material. Since this is a theoretical only paper, important proofs should be included.

Correctness: Overall the claims seem correct (although I did not check the math exhaustively)

Clarity: The paper is well written.

Relation to Prior Work: The paper clearly discusses prior work and places the contributions in context. However, I would prefer if table 1 from the supplementary material was included in the main paper, to provide the context of the contribution better.

Reproducibility: Yes

Additional Feedback: Overall a nice paper. I am satisfied with the reviewers responses and I recommend acceptance.

[Author Response · NeurIPS 2020]

We are very grateful to the reviewers for their helpful feedback and suggestions, and are pleased to have received a
generally positive response. Our responses to the main concerns are given as follows. All citations refer to the reference
list in the main document.

---

**[Responses to multiple reviewers] (Experimental evaluations)** We appreciate that experiments are important for
many papers, but do not believe it to be the case for this theory paper; please note that prominent related works such as
[8,23,24,37] also do not include experiments. At least in the special case of linear measurements, extensive numerical
results for (an approximation of) the $\mathcal{K}$-Lasso have already been presented in [2].

**(Novelty and Insight)** While our paper shares similar high-level insights to earlier works, in particular showing that
non-linear observations may be treated as noisy linear observations [23,28], we believe that the *direct* study of generative
priors adds significant value to existing approaches based on the Gaussian mean width (GMW) and related notions.

Our analysis builds on works such as [2,15,23,24], but we believe that these techniques are combined and extended in a
novel manner, with distinct proofs. For instance: (i) Compared to [23,24], we attain $m^{-\frac{1}{2}}$ scaling instead of $m^{-\frac{1}{4}}$, and
make no use of GMW throughout our main analysis; (ii) We require the careful control of several terms in (53), which
in turn requires proving Lemmas 7 and 8, along with a more involved chaining argument compared to [2,15]; (iii) We
address the uniform recovery case via the LEP, and are the first to do so to our knowledge.

---

**[Responses to R1]: (Assumptions)** The unit-norm assumption is standard in this line of works, and it (or similar) is
indeed essential in the 1-bit model. More generally, the generalized Lasso does not depend on $f(\cdot)$, so if $\mathbf{x}_0$ and $c\mathbf{x}_0$
are both feasible under the prior, one cannot distinguish $f(\langle \mathbf{a}, \mathbf{x}_0 \rangle)$ from $\tilde{f}(\langle \mathbf{a}, c\mathbf{x}_0 \rangle)$, where $\tilde{f}(z) = f(z/c)$. See also
Section 4.4 for a related discussion and generalizations to non-unit norms. Indeed, certain models such as phase retrieval
do not satisfy our sub-Gaussianity assumption, and we will better highlight this in the revision. This assumption is still
much more general than the 1-bit and linear models, and is adopted in many prior works including [9, 24, 27].

**(GMW)** We will make explicit that the GMW calculation assumes $G(B_2^k(r))$ is contained in the unit ball, and highlight
the limitation mentioned for a large radius $r$. We believe that the unit-ball setting remains of significant interest in itself.

**(Practicality in the 1-bit case)** In [15] it is assumed that the feasible set lies in the unit sphere, so it is fair to
assume the same for comparison. In more detail, [15, Corollary 3] gives a guarantee on any $\hat{\mathbf{x}}$ such that $\mathbf{A}\hat{\mathbf{x}}$ has
small Hamming distance to $\tilde{\mathbf{y}}$, but does not specify an optimization problem for finding such $\hat{\mathbf{x}}$. The problem
$\min_{\mathbf{x} \in \text{Range}(G)} \mathrm{d}_{\mathrm{H}}(\mathbf{A}\mathbf{x}, \tilde{\mathbf{y}})$ appears to be very hard to solve (e.g., being highly non-differentiable and combinatorial),
and the heuristic in [15, Section V] can be viewed as first approximating $\mathrm{d}_{\mathrm{H}}$ by a convex function, and then *further*
approximating the minimizer of that function. In contrast, the generalized Lasso solution can be approximated *directly*
using gradient methods. However, we acknowledge that both approaches still require some level of approximation, and
will accordingly significantly tone down and clarify the claim of practicality.

**(Corollary 1)** The assumption $\frac{\bar{\mu}G(\mathbf{z}^*)}{\|G(\mathbf{z}^*)\|_2} \in \mathcal{K}$ will be satisfied, for instance, when the generative model is a ReLU
network with no offsets (see [37, Remark 2.1]), due to $\mathcal{K}$ being cone-shaped. The sub-Gaussianity constant is indeed
dependent on $\mathbf{z}^*$, but it can be upper bounded independently of $\mathbf{z}^*$ in special cases of interest, including any model in
which the measurements are uniformly bounded (e.g., including not only 1-bit, but also more general multi-bit quantized
models). We will point out these examples, but also highlight that these assumptions may pose some limitations.

Despite the slight limitations of the GMW-based and NN-based results, we note that these are relatively minor corollaries,
and hope that the final decision is primarily based on our main theorems.

**(Flow of main body)** We would be happy to move some of the less central corollaries (e.g., Sections 4.2 and 4.5) to the
appendix for improved flow in the main body. We could use the extra space for additional intuition and/or brief outlines
for the main proofs, as suggested by R4.

---

**[Responses to R2]: (LEP in Assumption 1)** The intuition behind the LEP in Definition 2 is simply that if $\mathbf{x}_1$ is close
to $\mathbf{x}_2$, then $\tilde{f}(\tilde{\mathbf{A}}\mathbf{x}_1)$ is close to $\tilde{f}(\tilde{\mathbf{A}}\mathbf{x}_2)$. The statement of Assumption 1 is somewhat more cumbersome for technical
reasons (e.g., to make (114) rigorous), but we will aim to further highlight this intuition. Our paper includes formal
verifications of the LEP for the linear and 1-bit models, and we expect that further examples could be established, but
we defer this to future work. (Please see "Responses to multiple reviewers" above regarding the other comments.)

---

**[Response to R3]: (GMW)** Indeed, if the GMW is defined for $\mathcal{K}$ instead of $\mathcal{K} - \mathcal{K}$, the factor 2 can be omitted;
however, $\mathcal{K} - \mathcal{K}$ is more commonly used. In the revision, we will further highlight that $Lr = n^{\Omega(1)}$ is typical for neural
networks [2] after stating Theorem 1. We will also correct the typo in Line 202.

---

**[Response to R4]: (Proofs)** In our experience, having all proofs in the appendix is not uncommon for theory papers at
NeurIPS. However, we would be happy to include some additional proof intuition/outlines in the revision, using the 9th
page and/or some space freed up by moving Sections 4.2 and 4.5 to the appendix.

[Meta-Review · NeurIPS 2020]

This paper proposes a novel analysis for reconstruction of signals from nonlinear measurements using what's called K-Lasso, leveraging on generative priors. This is an important emerging area and this is a solid contribution. In our the updated reviews there are suggestions for improving the presentation, in particular section 4.